

# Do convection-permitting regional climate models have added value for hydroclimatic simulations? A test case over small and medium-sized catchments in Germany

Oakley Wagner[1,3], Verena Maleska[2], Laurens M. Bouwer[1,3]

[1]Climate Service Center Germany (GERICS), Helmholtz-Zentrum hereon, 20095 Hamburg, Germany
[2]Chair of Environmental Development and Risk Management, Technical University of Dresden, 01062 Dresden, Germany
[3]Institute of Geography, University of Hamburg, 20146 Hamburg, Germany

*Correspondence to*: Oakley Wagner (oakley.wagner@hereon.de)

**Abstract.** Through fine grid structure and explicit representation of deep convective processes, convection-permitting
regional climate models (CPRCMs) bear great potential for improved assessment of climate and hydrology under current and
future climatic conditions. For a robust assessment of the added value of CPRCMs as climate service for hydrological impact
modelling, the current scope of research needs to be expanded by studies on further model structures and study areas. The
paper presented here considers the non-hydrostatic model ICON-CLM 2.6.4 at 3 km resolution (ICON3km) and its driving
model ICON-CLM 2.6.4 with parametrised convection at 11 km (ICON11km) for a study area of 13,210 km² in East Central
Germany, enclosing the small (107 km²) to medium-sized (529 km²) catchments of the upper and central part of the
predominantly rural Weiße Elster river basin. The reanalysis-driven historical hourly air temperature, global radiation,
relative humidity, wind speed and precipitation simulations are evaluated. ICON3km is further analysed for added value for
discharge simulations using the distributed hydrological model WaSiM. Our results suggest primarily an improvement by
ICON3km in the estimation of summer air temperature and global radiation, as well as reducing the overestimation of the
left tail of the frequency distribution of wind speed. The most noticeable deficiencies of ICON3km are the strong
overestimation of high precipitation intensity and too frequent heavy rainfall events. These shortcomings translate into a
pronounced overestimation of discharge when uncorrected, dominating the hydrological estimations. As such, no added
value in the use of ICON3km for hydrological impact modelling in the Weiße Elster basin was identified.

## 1 Introduction

Extreme rainfall events around the world are expected to become more intense and more frequent under global warming
(IPCC, 2021; Myhre et al. 2019). In the UK for example, the frequency of rainfall events exceeding 20 mm/h, and thereby
capable of producing flash floods, is estimated to be 4-fold by 2070 under a high emission scenario (Kendon et al., 2023).
With urbanisation on the rise worldwide (Jiang & O'Neill, 2017) and a growth in population in flood-prone areas (Kam et
al., 2021), an increase in the frequency and intensity of local extreme rainfall events leads to large flood risks. Convection-





permitting regional climate models (CPRCMs) bear great potential for improved projections of changes in extreme rainfall. These models operate on spatial resolutions smaller than 4 km (Prein et al., 2021) and are built with a non-hydrostatic dynamical core, since the hydrostatic approximation is no longer valid at this scale (Giorgi, 2019; Steppeler et al., 2003). As such, they resolve deep convection and no longer rely on its parametrisation, thereby circumventing a major source of uncertainty (Prein et al., 2015). However, finer scale processes, such as cloud and precipitation microphysics, shallow

convection, radiative transfer, and turbulence still need to be parametrised (Prein et al., 2015; Kendon et al., 2021). A full depiction of all energetic processes in clouds would require a downscaling to the smallest scale, i.e. the Kolmogorov scale, where viscous dissipation converts turbulent kinetic energy into heat (Prein et al., 2015; Friedlander and Topper, 1961 cited in Schulz & Sanderson, 2004). Besides a more process-based representation of atmospheric processes, CPRCMs also introduce finer grid spacing, allowing to better capture complex topography and land surface heterogeneities (Gutowski et

al., 2020), constituting an added value for impact modelling over small catchments of mountainous or highly urban character (e.g. Schaller et al., 2020; Tamm et al., 2023).

To justify their high computational costs and data storage requirements (Schär et al., 2020), CPRCMs must prove to indeed offer substantial improvement compared to coarser regional climate models with parametrised convection (RCMs). However, in the field of hydrological impact assessment, to this day only a limited number of studies comparing the use of

CPRCMs to RCMs have been conducted. An overview is given in Table 1, reflecting how from a state of unsatisfactory performance for impact modelling in the early stages of development (e.g. Kay et al., 2015; Reszler et al., 2018), CPRCMs have made remarkable progress and come to offer substantial added value for flood simulation. However, with most studies only focusing on a single basin, using data from a single CPRCM, over a short simulation time, there remains an urgent need for further studies (Lucas-Picher et al., 2021). Our paper introduces an additional geographical region and CPRCM to the

discourse. It looks at the strengths and limitations of ICON-CLM (2.6.4) in convection-permitting setup at 3 km resolution in depicting hourly near-surface air temperature and relative humidity, wind speed at 10 m, global radiation and precipitation over a chosen catchment in East Central Germany, compared to its driving coarser RCM, ICON-CLM (2.6.4) at 11 km resolution. The climate model data is further used as input to the distributed hydrological model WaSiM to identify potential added value of the studied CPRCM data for discharge simulations in the catchment.

**Table 1: Previous studies investigating the added value of CPRCMs compared to RCMs, with non-bias-corrected (*) and bias-corrected (+) precipitation data, for hydrological impact modelling (exclusively model chains consisting of a climate or numerical weather prediction model and a hydrological model are considered)**

| Study | Study Region | Driving GCM/ RCM | CPRCM | Hydrological Model | Key Findings |
|---|---|---|---|---|---|
| Kay et al. (2015) | southern Britain | limited-area version of HadGEM3 (12 km) | modified version of the UK Met Office weather forecast model | CLASSIC-GB (semi-distributed) | - Worse performance for hydrological simulations using CPRCM data due to its strong |



| | | | (resolution: 1.5 km) * | | - positive bias in precipitation intensity |
|---|---|---|---|---|---|
| Mendoza et al. (2016) | Colorado River basin, south-western USA | WRF (12 km, 36 km) | WRF (4 km) * | PRMS (distributed), VIC (semi-distributed), Noah LSM (semi-distributed), Noah-MP LSM (semi-distributed) | - Improvement in the simulation of historical runoff ratios by using CPRCM data |
| Reszler et al. (2018) | south-eastern Austria | COSMO-CLM, WRF (12.5 km, 50 km) | COSMO-CLM, WRF (3 km) *, + (scaled distribution mapping) | KAMPUS (distributed) | - CPRCM data was not found to offer added value for hydrological modelling |
| Schaller et al. (2020) | West Coast of Norway | EC-Earth v2.3 (25 km) | AROME-MetCoOp (2.5 km) * | Hydrologiska Byråns Vattenbalans (semi-distributed and distributed) | - spatial refinement and higher temporal resolution advantageous in the modelling of fast flood-generating processes in complex terrain |
| Rudd et al. (2020) | southern Britain | Met Office Hadley Centre model (12 km) | Met Office Hadley Centre model (1.5 km) * | Grid-to-Grid (distributed) | - finer grid spacing advantageous in exceedance count projections, especially for minimal and minor property severity impact level |
| Kay (2022) | Britain | HadREM3-GA705 (12 km) | HadREM3-RA11M (2.2 km) *, + (monthly factors) | G2G (distributed) | - added value by CPRCM in respect to percentage bias in low flow volume, median flow and high flow volume for baseline simulations<br>- divergences between use of CPRCM and RCM data mostly abolished with bias-correction |
| Davis et al. (2022) | Bangalore catchment, South India | Global Forecasting | WRF (3km, 1 km; 15-min | PCSWMM (semi-distributed) | - improvement in the simulation of urban flood level peaks |



| | | System (25 km; 3-hour intervals) | intervals) * | | using CPRCM data |
|---|---|---|---|---|---|
| Tamm et al. (2023) | Southern Finland | HARMONIE (12 km) | HARMONIE (3 km) + (empirical quantile mapping) | SWMM (distributed) | - more realistic annual maximum flow values using CPRCM data |
| Ascott et al. (2023) | Lake Victoria basin, Eastern Africa | Met Office Unified Model (25 km) | Met Office Unified Model (4 km) * | GR4J (lumped) | - higher flow estimates using CPRCM data |
| Poncet et al. (2024) | Gardon d'Anduze catchment, southern France | CNRM-ALADIN (12 km) | CNRM-AROME (2.5 km) *, + (CDF-t method) | GR5H (lumped), CREST (distributed) | - improved simulation of flood intensity and frequency using the uncorrected CPRCM data<br>- most intense floods best modelled with bias-corrected CPRCM data compared to RCM data, but RCM data served for better depiction of the distribution of flow peaks over threshold |

In this paper, section 2 outlines the study area considered in this paper, the observational and climate model data used, as well as the applied methodology. The results from the meteorological and hydrological data analysis are presented in section

3, while section 4 provides a discussion. A conclusion is given in section 5.

## 2   Methods and Data

### 2.1   Study Area

The study area was chosen to encompass the focus regions of the KlimaKonform project (Zorn et al., 2024) in East Central Germany (Fig. 1a). It spans 13,210 km² and hosts the upper and central part of the Weiße Elster river basin (Fig. 1b). The

twelve enclosed catchments stretch over an area of 2,960 km², with the smallest having an area of 107 km² (Dröda) and the largest of 529 km² (Strassberg).



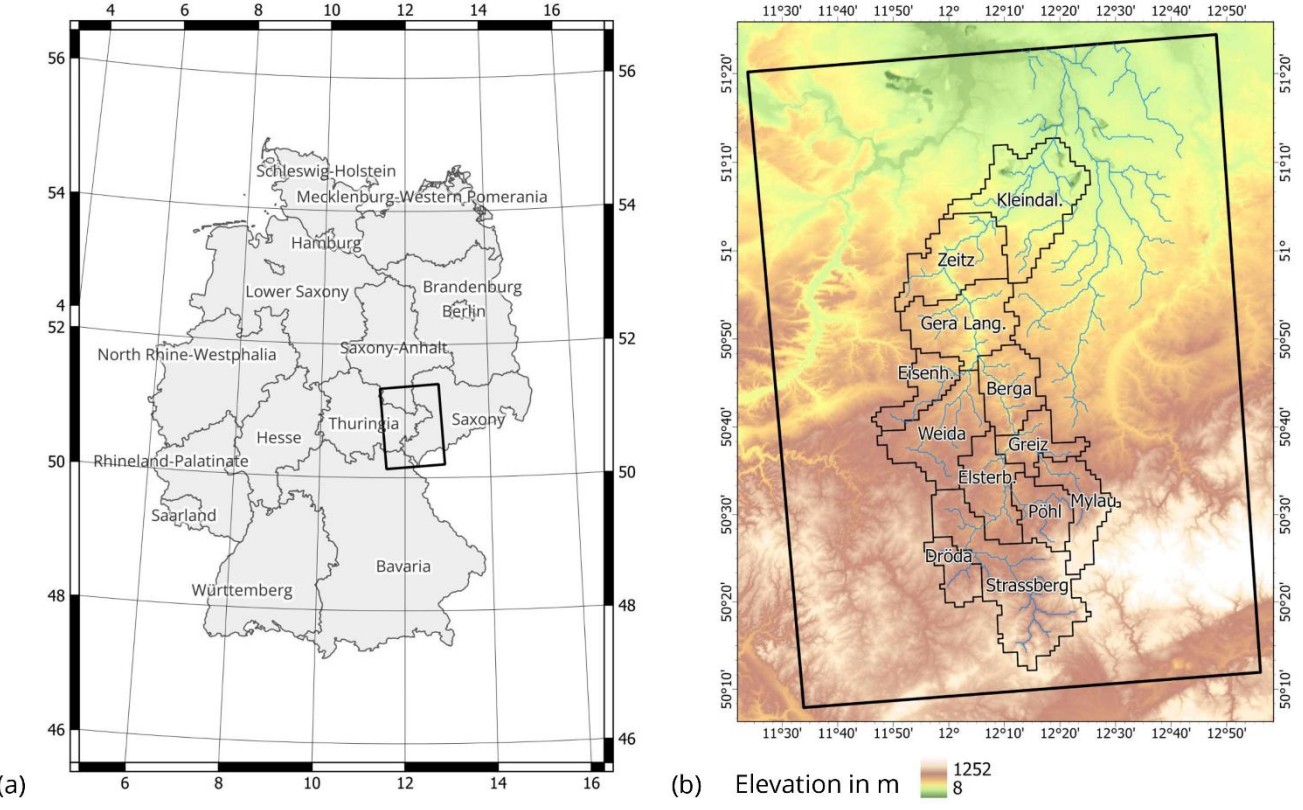

**Fig. 1: (a) Location of the study area in Germany and (b) the studied catchments within the area over a digital elevation model of 30 m resolution (SRTM 1 Arc-Second Global; USGS, 2018)**

The study area is embedded between the Leipzig lowlands to the North and the Elster Mountains and Ore Mountains to the South, resulting in a topographically varied landscape encompassing both flat plains and steeply sloped terrain (Rode et al., 2008; Miera et al., 2022). Altitudes range from 8 m to 1,252 m on a 30 m grid, from 85 m to 1,006 m on a 3 km grid, and from 101 m to 852 m on an 11 km grid. The land use in the Weiße Elster river basin is predominantly cropland and pasture (Rode et al., 2008, see Fig. S1 in the appendix). Chernozem/phaeozem and luvisol soils on loess are prevalent in the northern part of the study area, while cambisols and stagnic gleysols dominate the southern part (Miera et al., 2022).



## 2.2 Observational Data

Hourly observational data from 20 stations for mean air temperature, two stations for global radiation, 20 stations for relative humidity at 2 m, and 11 stations for wind speed at 10 m has been obtained from the German Weather Service (DWD) for the
period of 2005 to 2014 (Fig. S2 in the appendix). No spatial interpolation has been performed; the station measurements are compared point-based to the corresponding climate model grid cells. For the assessment of precipitation anomalies, 60-year time series of monthly point-based precipitation data (1963 - 2022) have been obtained from DWD for five stations in the study area.

For other evaluations of precipitation simulations and hydrological modelling, adjusted radar-based quantitative precipitation
estimations are used as reference, provided by DWD over Germany (RADOLAN product; Weigl & Winterrath, 2010) for the study period of 2005 to 2014 in hourly time steps at a resolution of 1 km. The data has been upscaled and regridded to the climate model grids using the area-weighted average method (becoming RADOLAN3km and RADOLAN11km). Kreklow et al. (2020) evaluated the performance of the original RADOLAN dataset against rain gauge data and found RADOLAN to overestimate the number of days with daily precipitation sums > 1 mm, as well as heavy rainfall days (i.e. daily precipitation
sums > 20 mm), in the summer half-year. However, the average daily precipitation sum for these respective days was underestimated (Kreklow et al., 2020). Similar trends were reported for the winter half-year, yet far less pronounced (Kreklow et al., 2020). Especially up to the year 2010, extreme high outliers were not uncommon in the time series, with markable improvements through the introduction of additional gauges in 2007, as well as further quality checks and new processing routines in 2010 (Kreklow et al., 2020).

Discharge data along the Weiße Elster has been obtained through the ReKIS project (Kronenberg et al., 2021), based on measurements by the Sächsisches Landesamt für Umwelt, Landwirtschaft und Geologie (LfULG), Thüringer Landesamt für Umwelt, Bergbau und Naturschutz (TLUBN) and Landesamt für Umweltschutz Sachsen-Anhalt (LAU).

Topographic information was obtained from SRTM 1 Arc-Second Global (USGS, 2018), land use is based on Corine Land Cover 2018 (Copernicus Land Monitoring Service, 2020) and used soil data refers to BÜK200 (BGR & SGD, 2018).

## 100   2.3   Climate Model Data

The regional climate model ICON-CLM (version 2.6.4, Pham et al., 2021) has been run over the EURO-CORDEX domain at 11 km resolution for hourly output (ICON11km). It is driven with data from the global atmospheric reanalysis ECMWF-ERA5 and provides the lateral boundary conditions for the CPRCM. As CPRCM, ICON-CLM (version 2.6.4) is employed in its convection-permitting setup at 3 km resolution over the Central European (CEU) domain providing hourly weather data
(ICON3km). The domains are illustrated in Fig. S3 in the appendix. The considered time series span from 2005 to 2014.



## 2.4 Evaluation of Simulated Meteorological Data

Air temperature, global radiation, relative humidity, and wind speed station measurements are compared to their co-located climate model grid cells. Precipitation simulations are evaluated raster-based against radar precipitation estimations upscaled and regridded to the climate models' grids. Comparing precipitation simulations to observations on the same grid circumvents the need to consider differences in the degree of smoothing of local heavy rainfall fields with resolution (Iles et al., 2020). While observations have been upscaled to the respective climate model grids, it has been retained from harmonising the climate models' grid spacings. Upscaling the finer climate simulation results to the grid of the coarser climate model would entail partial information loss (Iles et al., 2020), while downscaling the coarser climate model simulations would bring them into resolutions where processes dominate which are on a scale too fine to have been computed by the coarser model (Prein et al., 2016). On grids of different resolution, a direct comparison between the climate model results is however not reasonable. It is worth noting that first studies suggest that improvements at higher resolution may still be apparent after upscaling (e.g. Prein et al., 2016; Fantini et al., 2018 for upscaling from ~ 12 km to ~ 50 km resolution).

When temporally aggregating observational data, time periods are not summarised if more than 20% of their data is missing (Haylock et al., 2008). Smaller gaps are filled by dividing the totals through the proportion of non-missing observations (Haylock et al., 2008). Frequency analyses for the 99.5th-percentile of hourly precipitation have been conducted based on frequency polygons with a number of bins according to Sturges' rule (Sturges, 1926). The quantile thresholds are deduced from RADOLAN data for both resolutions and transferred to the respective ICON data sets.

Depth-duration-frequency curves have been constructed by sliding a moving window of a fixed duration (2 h, 4 h to 24 h by 2-hour-increments for 11-km-resolution data, and by 4-hour- increments for 3-km-resolution data) over the time series and fitting the identified annual-maximum precipitation sums with a Gumbel distribution (Koutsoyiannis et al., 1998). The short duration of the time series of only 10 years poses a strong limitation to this approach.

## 2.5 Hydrological Modelling

The distributed, physically based deterministic hydrological model WaSiM (Message Passing Interface, model version: Richards 10.00.03; see Schulla, 2021) has been run for hourly time steps for the Weiße Elster river for the study period of 2005 to 2016. The first simulation year has been discarded to ensure adequate spin-up. Meteorological input variables consist of hourly air temperature, global radiation, relative humidity, wind speed and precipitation. Topography, land use/ land cover (LULC) and soil characteristics have been embedded on a 1 km² grid for all simulation runs. The vertical water movement through the unsaturated zone was described by the Richards equation (Richards, 1931) and potential evapotranspiration was calculated according to the approach after Penman-Monteith (Monteith, 1975; Brutsaert, 1982). As interpolation technique for the meteorological input data, Thiessen polygons were chosen in order to keep the grid structure of the climate model



data unchanged. Radiation and temperature were adjusted considering the impact of topography, according to the implemented scheme by Oke (1987).

## 3 Results

### 3.1 Evaluation of Simulated Meteorological Data

#### 3.1.1 Near-Surface Air Temperature

According to a Welch two sample t-test, both the RCM and the CPRCM compute significantly lower hourly air temperatures than have been observed on average over the study period ($\overline{t}_{Obs}$ = 8.7°C, $\overline{t}_{ICON3km}$ = 8.4°C, $\overline{t}_{ICON11km}$ = 8.3°C) at 95% confidence level. However, these cold mean biases fall within the measurement uncertainty bandwidth of the temperature stations of ± 0.08 - 0.76 K (Brinckmann & Dirksen, 2020). As such, no statement of added value by the CPRCM can be made for mean hourly air temperature simulation. It is however noticeable that the temperature estimates by the RCM are statistically significantly different from those by the CPRCM.

In the frequency distribution of hourly air temperature and in the representation of its diurnal cycle, ICON3km is not found to offer noticeable improvement compared to ICON11km.

While any identified improvement in the depiction of the monthly variability of air temperature blends in with the measurement uncertainty (Fig. 2), the results however hint towards (1) a greatest improvement by ICON3km compared to ICON11km in the summer months of June to August, and (2) a lower spatial bias variability throughout the year for ICON3km compared to ICON11km as reflected by the 95% confidence interval from the station means. Monthly mean bias errors of hourly air temperature were found to be greater in winter (DJF) than in summer (JJA) for both ICON3km and ICON11km.



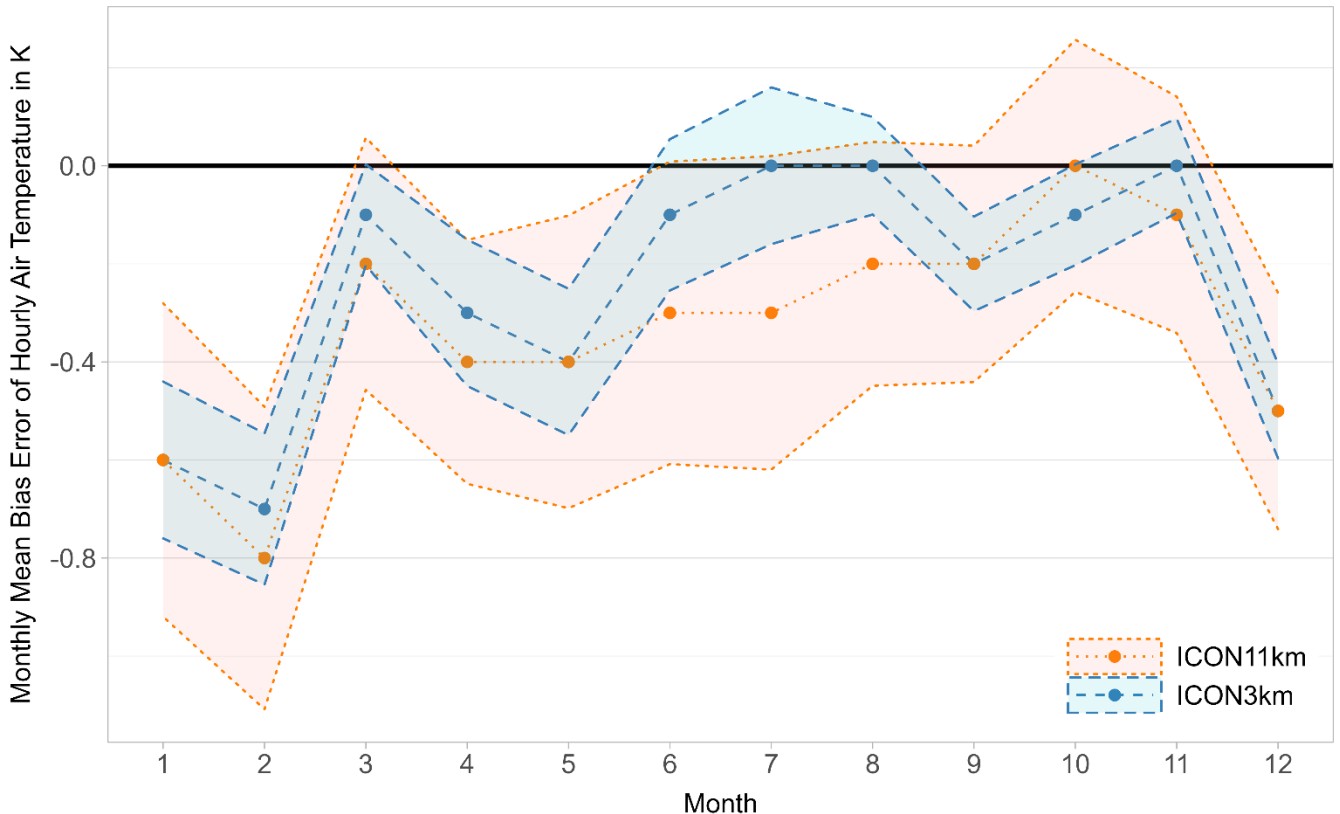

**Fig. 2: Monthly mean bias error of hourly air temperature of ICON3km and ICON11km during the period 2005 to 2014, as well as the 95%-confidence intervals from the station means**

### 3.1.2 Global Radiation

While both ICON3km and ICON11km compute daily mean global radiation significantly higher than observed according to a Welch two sample t-test ($\overline{G}_{ICON3km}$ = 45.9 J/cm², $\overline{G}_{ICON11km}$ = 45.9 J/cm², $\overline{G}_{Obs}$ = 44.5 J/cm²), however if the observations are assumed systematically too low in the order of the pyranometers' uncertainty of 3% (DWD, n.d.), the apparent overestimation by the climate models is not significant any more. Indeed, pyranometers face the challenge of temperature differences between their inner dome and detector, caused by infrared radiation from the instrument optics in conjunction

with different thermal capacities and connectivities (Philipona, 2002), and further intensified by differential cooling by the colder sky, especially on cloud-free days (Sanchez et al., 2015). These effects lead to thermal negative offsets if not controlled sufficiently well, and hence to an underestimation of global radiation by the pyranometers (Philipona, 2002; Sanchez et al., 2015). Daily global radiation estimates by ICON3km were also not found to be significantly different from those computed by ICON11km.





Both climate models were found to overestimate the frequency of daily mean global radiation for the range of bins between 45.7 and 106.7 J/cm² but underestimate the frequency of higher global radiation. Not only in their frequency, but also in their intensity, moderate daily mean global radiation observations were overestimated, while extremes were underestimated, by both ICON3km and ICON11km.

For most months of the year, the studied CPRCM does not offer noticeable improvement in the estimation of daily mean
global radiation, apart during the peak of summer (July) where ICON3km was found to reduce the negative monthly mean bias error of daily mean global radiation by 2.5 J/cm².

When looking at the diurnal cycle of global radiation, estimates from the climate models peak one hour earlier than the observations, however they are able to adequately represent the amplitude of the diurnal cycle (Fig. 3). No apparent improvement is offered by ICON3km compared to ICON11km.

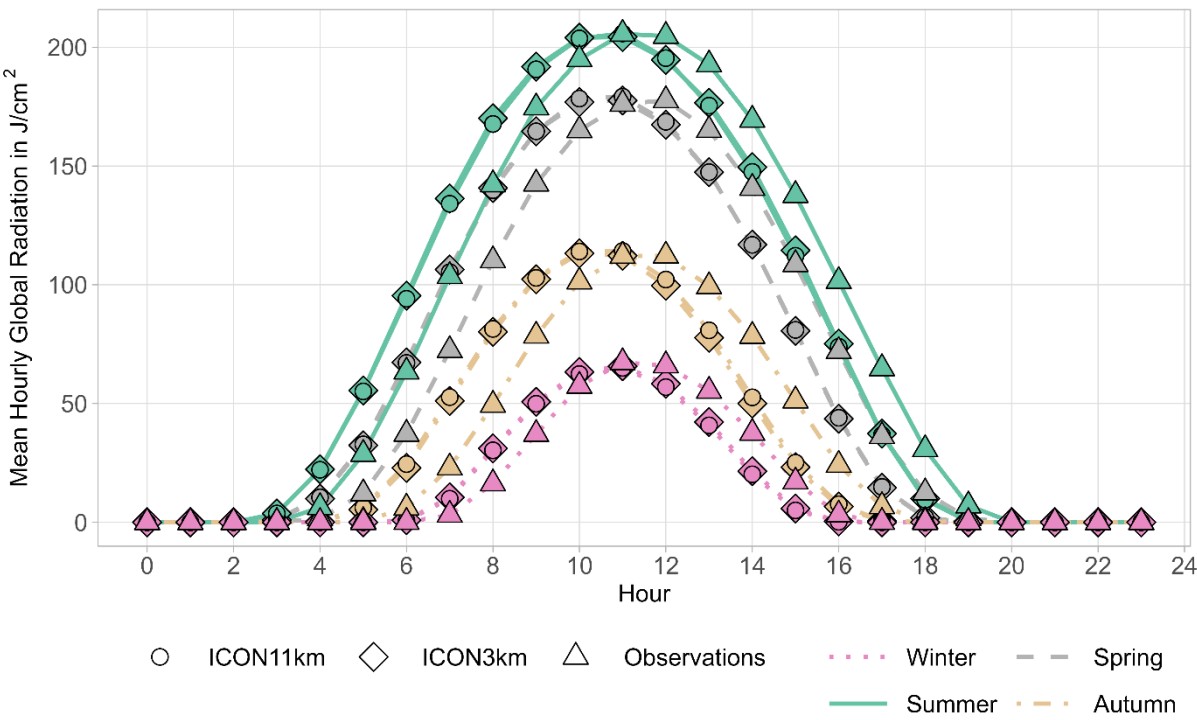

**Fig. 3: Seasonal diurnal cycles of hourly mean global radiation computed by ICON11km and ICON3km, as well as the observations for the study area over the time period of 2005 to 2014**

### 3.1.3  Relative Humidity

While hourly relative humidity computed by ICON3km is significantly higher than observed ($\overline{RH}_{ICON3km}$ = 78.7% vs $\overline{RH}_{Obs}$
= 78.3 %), this is not the case for the estimates by ICON11km according to a Welsh two sample t-test (with $\overline{RH}_{ICON11km}$ =





78.3%). Under consideration of the measurement uncertainty of ± 2 % of the employed sensor HMP45D (Kyrouac & Theisen, 2017), however neither of the climate models show significant differences to the observations. The estimates by the climate models were found to be statistically significantly different from each other.

ICON11km tends to underestimate the frequency bins of relative humidity of below 46.0 % and of above 89.8 %, while
overestimating those between 59.1 % and 89.8 %. ICON3km does not seem to offer noticeable improvement in the frequency distribution.

Overall, for most months ICON3km was found to outperform ICON11km in the estimation of hourly relative humidity. Only in the months of spring (MAM) was ICON11km found to offer lower monthly mean bias errors of hourly relative humidity compared to ICON3km. However, the annual maximal absolute difference between the mean monthly bias errors of
ICON3km and those of ICON11km is of only 1.3 % relative humidity, and thereby falls within the measurement uncertainty bandwidth. Both climate models tend to show negative relative humidity biases during winter (DJF) and autumn (SON), and positive biases during spring (MAM) and summer (JJA).

### 3.1.4 Wind Speed

According to a Welch two sample t-test, the climate models show significantly higher mean hourly wind speeds than have
been observed ($\overline{v}_{ICON3km}$ = 3.4 m/s, $\overline{v}_{ICON11km}$ = 3.2 m/s, $\overline{v}_{Obs}$ = 3.1 m/s). If the observations are assumed to be systematically too low in the order of the measurement uncertainty, the estimated mean from ICON11km is not significantly different from that of the observations. However, computed wind speeds by ICON3km remain significantly too high in the mean even under consideration of the measurement uncertainty. While this might suggest a performance decline by the CPRCM, it was however found to be the result of an improved representation of the frequency distribution (Fig. 4). Both
ICON3km and ICON11km tend to overestimate the frequency of moderate to high wind speeds (2.8 to 11.9 m/s, resp. 4.0 to 8.5 m/s), but ICON11km also does so for very low wind speeds (0.6 to 1.7 m/s), thereby attenuating its positive bias of the mean. While the climate models overestimate the frequency of high wind speeds, they underestimate their intensity, with slight improvements by ICON3km.



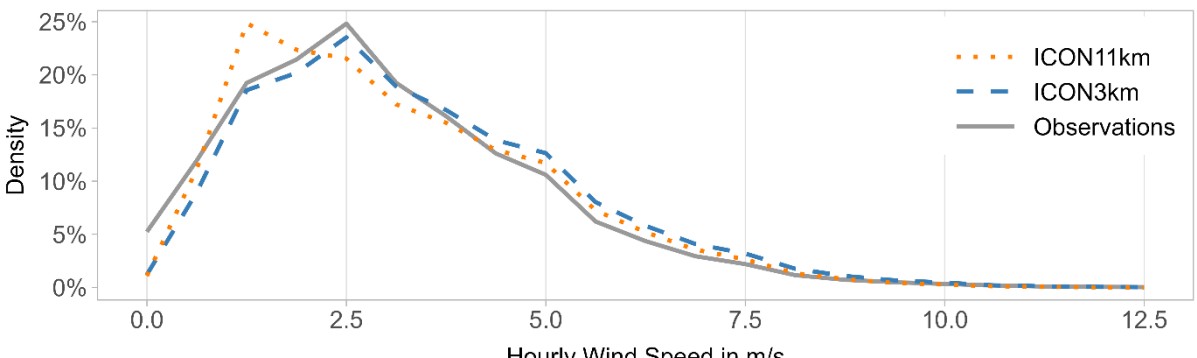

**Fig. 4: Frequency polygons for hourly wind speed as observed, and calculated by ICON3km and ICON11km over the study area for the period of 2005 to 2014**

ICON11km was found to compute a too pronounced amplitude of the diurnal cycle of hourly wind speed compared to the observations (Fig. 5). ICON3km shows a better performance yet tends to overestimate absolute values. While the observations plateau during the early afternoon, mean wind speed computed by ICON11km drops sharply after having reached its peak around noon, together with solar radiation. Mean wind speed simulations by ICON3km do not show a peak as sharp, but level out shortly, and are thereby in better alignment with the observations.

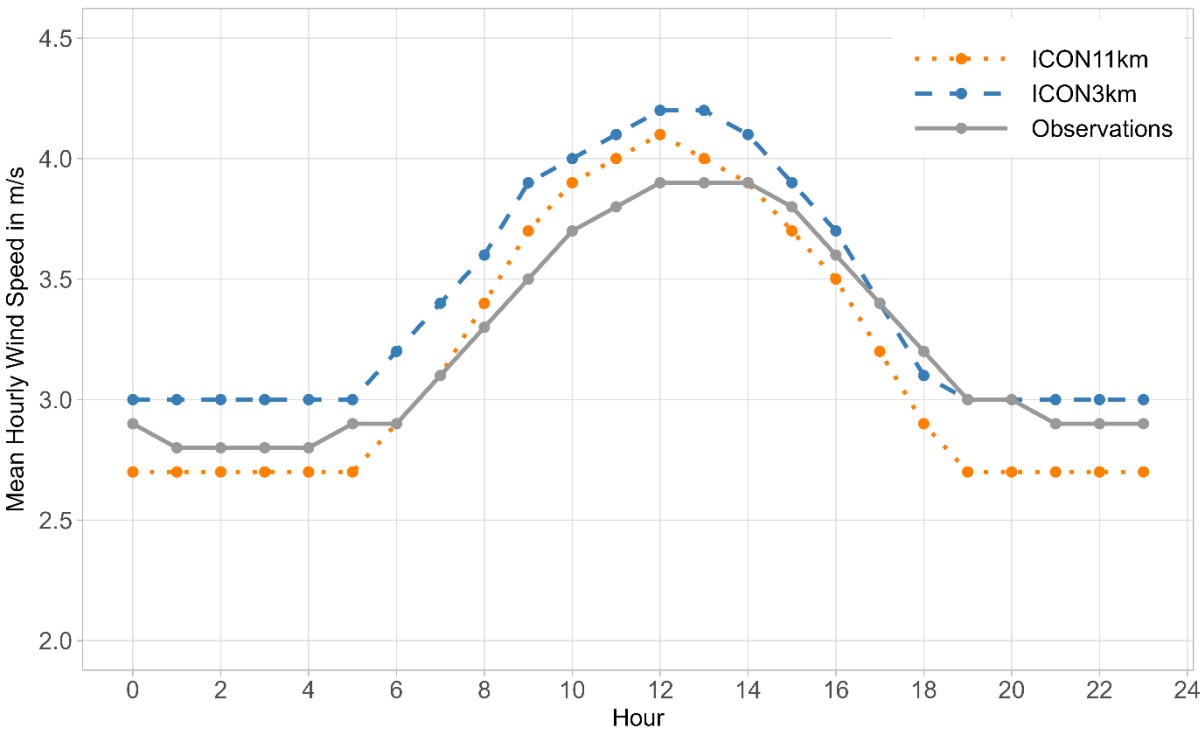

**Fig. 5: Diurnal cycle of mean hourly wind speed computed by ICON11km and ICON3km, as well as the observations for the study area over the time period of 2005 to 2014**



### 3.1.5 Precipitation

With median hourly precipitation of 2.8 mm over the 99.5th percentile (left-closed interval), ICON11km shows an underestimation compared to adjusted radar data upscaled to the same grid (RADOLAN11km) with a respective estimate of 3.8 mm. In turn, ICON3km overestimates the median of the 99.5th percentile of hourly precipitation compared to upscaled adjusted radar estimates (RADOLAN3km), with 4.2 mm to 4.0 mm respectively. These trends are further visible in the QQ-plots (Fig. 6).

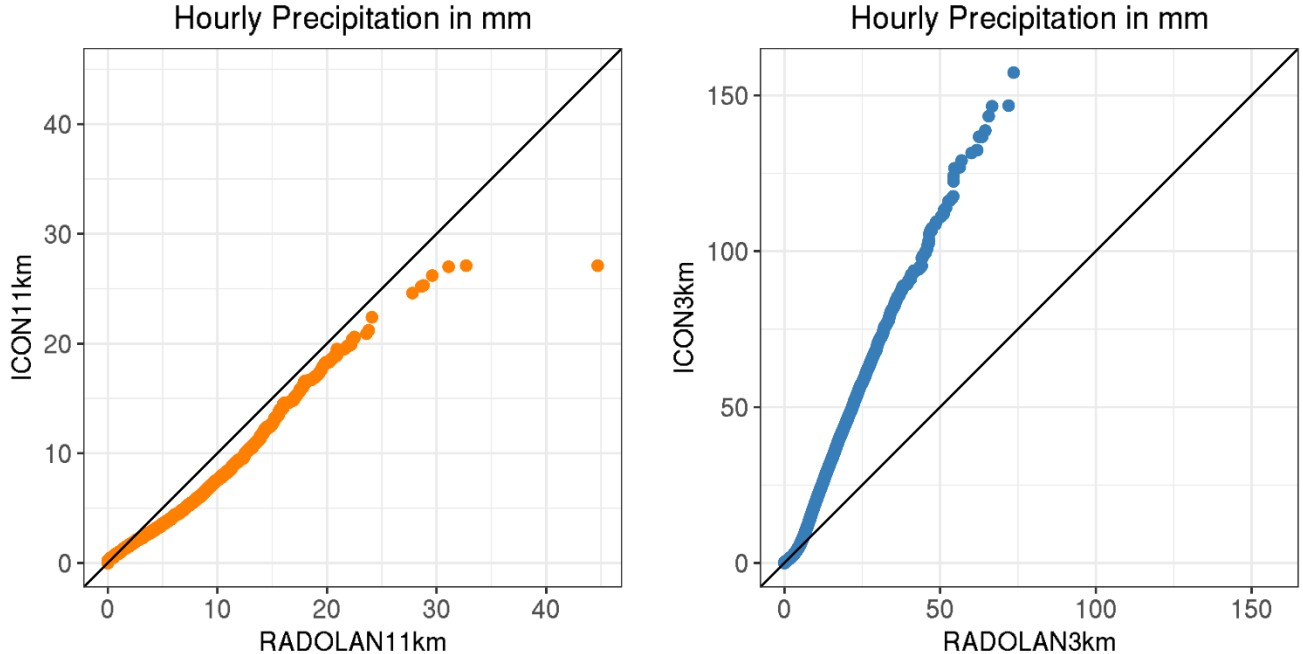

**Fig. 6: QQ-plots for hourly precipitation over the period of 2005 to 2014 for ICON11km to RADOLAN11km (left) and ICON3km to RADOLAN3km (right)**

The frequency polygons for the 99.5th-percentiles of hourly precipitation, with the thresholds drawn from the RADOLAN data for both resolutions, show ICON11km to compute too frequent low intensity rainfall, while ICON3km gives an underestimation (Fig. 7). The frequency of moderate and heavy rainfall in turn is underestimated by ICON11km but overestimated by ICON3km.





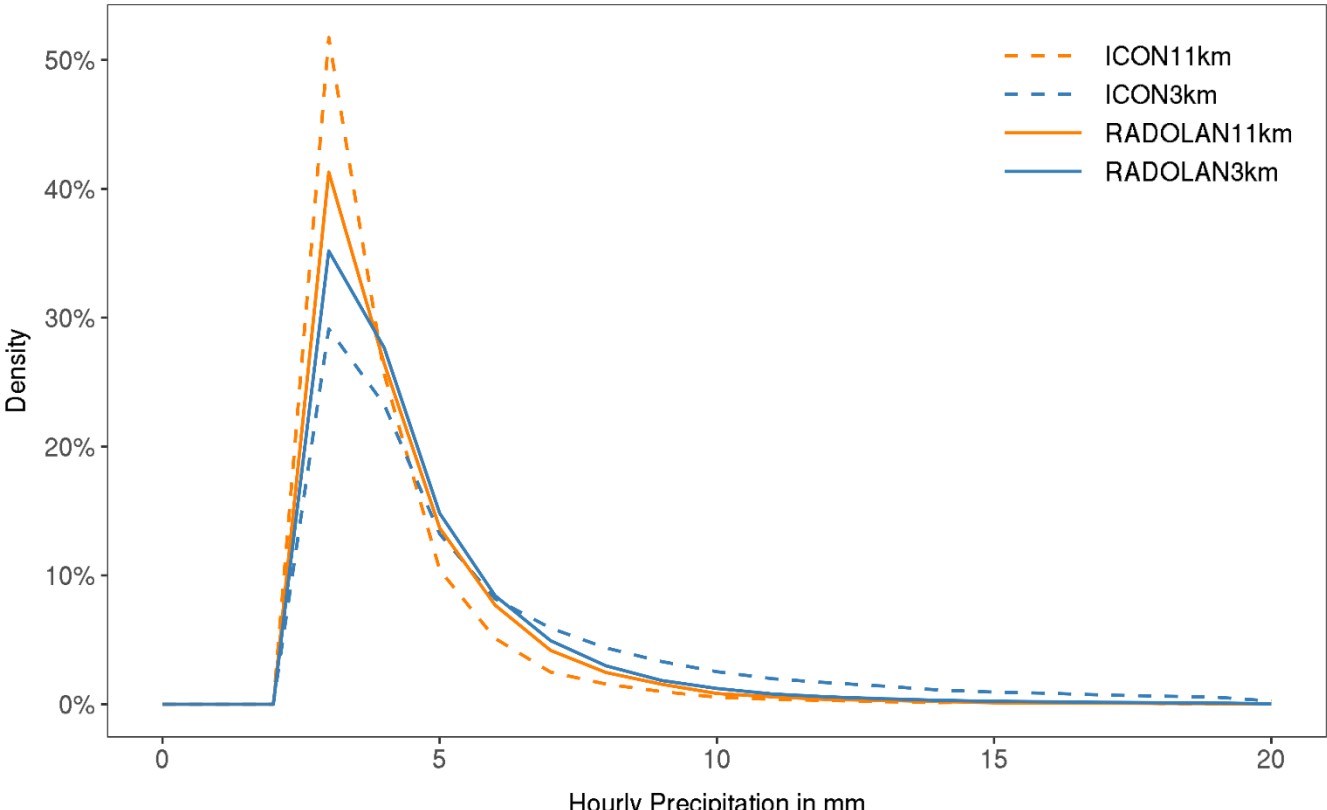

**Fig. 7: Frequency polygons for ICON11km, ICON3km, RADOLAN11km and RADOLAN3km for the 99.5th-percentile of hourly precipitation for the period of 2005 to 2014**

The overestimation of high precipitation intensities and their frequencies by ICON3km further results in too high depth-duration-frequency curves, as shown for selected return periods in Fig. 8. In keeping is an overestimation of the mean yearly precipitation totals by ICON3km compared to the radar data upscaled to equal resolution (790 mm compared to 670 mm). An overestimation of the frequency of light precipitation by ICON11km in turn also results in too high mean yearly precipitation totals by ICON11km compared to adjusted radar data estimates of respective resolution (729 mm compared to 665 mm).



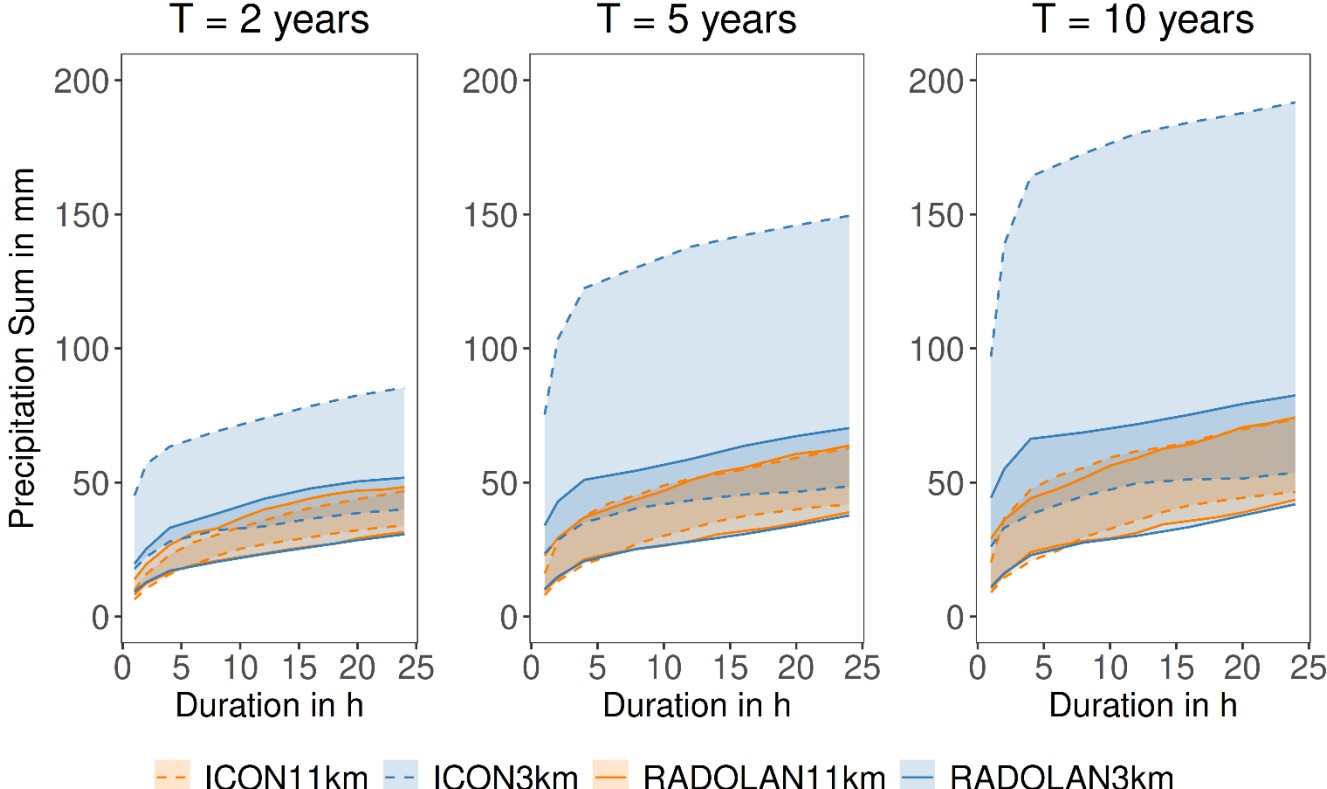

**Fig. 8: Minimum-maximum bandwidth of the depth-duration-frequency curves from all cells by ICON11km and ICON3km, together with the corresponding estimates from upscaled adjusted radar data (RADOLAN11km and RADOLAN3km) for return periods T of 2, 5 and 10 years**

The monthly means of the deviations of daily precipitation computed by ICON3km and ICON11km to the adjusted radar data estimates upscaled to their respective resolution (Fig. S4 in the appendix) are similar between the climate models during autumn (SON) and winter (DJF) but differ strongly during the peak of summer. In fact, in the summer month of July, the monthly mean bias error of daily precipitation is three times as high for ICON3km than for ICON11km (0.9 mm to 0.3 mm), given the overestimation of heavy precipitation by ICON3km. In no season was ICON3km found to offer monthly mean daily precipitation biases lower than ICON11km. However, the discussed mean bias shows a narrower range among the individual climate cells in ICON3km than in ICON11km, as reflected by their 95%-confidence interval, for all four seasons.

Furthermore, through finer grid spacing, ICON3km allows for a better spatial delineation of local precipitation fields, as are particularly distinct over mountainous regions, such as the Ore Mountains in the south-east of the study area. Additionally, smaller grid cells avoid that local heavy rainfall events occurring outside of a catchment are channelled into the latter in the frame of impact modelling through joint overlap of the simulated precipitation field and the catchment by a coarse climate model cell.





Positive monthly precipitation anomalies to a long-term mean were found to be predominantly overestimated by the climate models (Fig. 9), especially by ICON3km, as has also been seen reflected through the overestimation of the mean yearly

precipitation sums. However, the climate models were still found to be able to reach into the deep negative monthly precipitation anomalies, sometimes even overestimating them. For a considerable number of months, the negative anomalies remain however underestimated, making the picture not as clear cut as for the depiction of positive precipitation anomalies. The same holds true when it comes to the performance comparison of ICON3km and ICON11km.

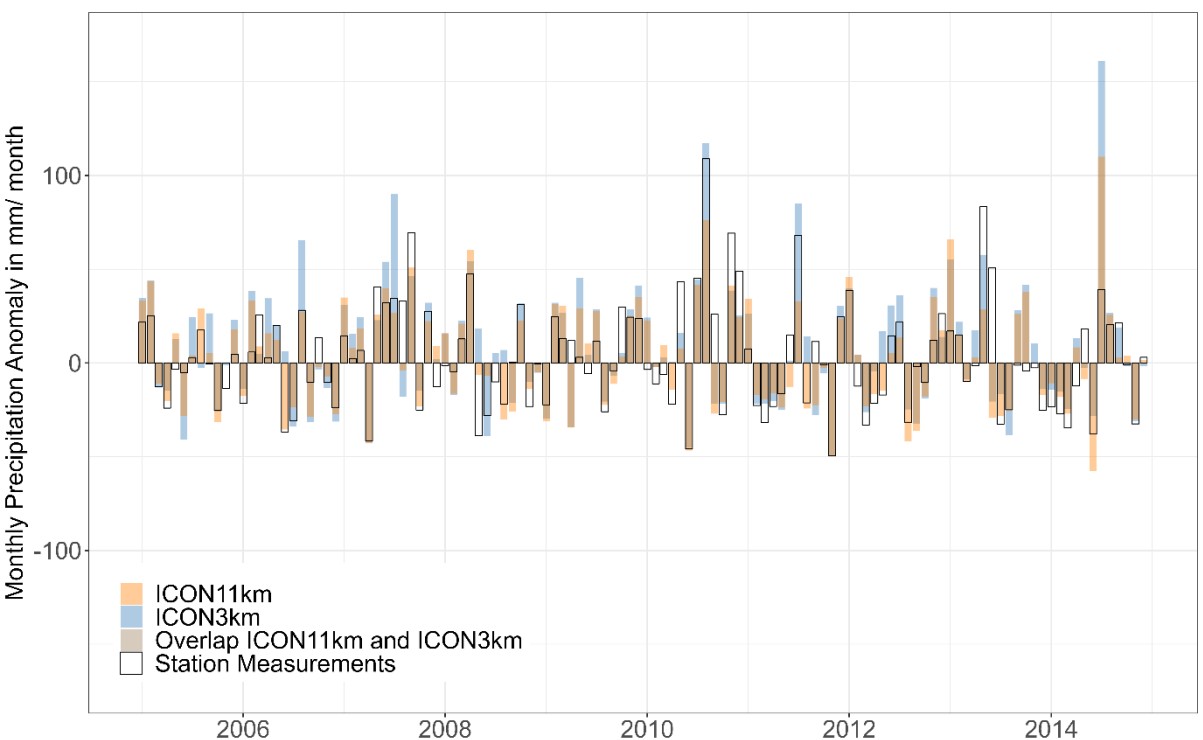

**Fig. 9: Monthly precipitation anomalies to the long-term reference (1963 – 2022) for the years 2005 to 2014 for ICON11km, ICON3km and the station measurements**

## 3.2 Hydrological Analysis

### 3.2.1 Calibration and Validation of the Hydrological Model

The hydrological model was calibrated on the largest summer flood in the area during the study period (flood of June 2013),

looking at the seven catchments for which discharge observations were available, with Nash-Sutcliffe efficiency (NSE) values ranging between 0.63 and 0.95 and Kling-Gupta efficiency (KGE) values between 0.76 and 0.96 for six of these catchments, while the catchment of Weida constitutes an outlier with a NSE of only 0.14 and a KGE of 0.60. The simulated responses generally fell short on depicting the flashiness of the catchments and all but two of them (viz. Eisenhammer and Zeitz) showed a retardation of the flood peak. Furthermore, the baseflow was found to be erroneously modelled as barely




responsive. The model was validated on a winter flood, given that no other pronounced summer flood occurred during the study period. The model was found to greatly overestimate direct discharge from snowmelt, computing too high and retarded flood peaks, as well as too flashy hydrographs, with a persisting underestimation of the baseflow response.

Looking at weekly discharges over selected years for calibration and validation, the model results were too high in direct discharge, interflow and baseflow.

Assuming the simulated discharges to be systematically distorted by the hydrological model, a qualitative comparison between the concurrent hydrological responses computed based on ICON and RADOLAN data is still possible.

### 3.2.2 Continuous Simulations

The hydrological model driven with meteorological data from ICON11km was found to compute median hourly routed discharge values over the study period of 2006 to 2014 that are lower than when driven with adjusted radar data of the same

resolution (RADOLAN11km) (Fig. 10). In turn, when driven with data from ICON3km, the hydrological model simulated median hourly routed discharges that were higher than estimates based on adjusted radar data of equal resolution (RADOLAN3km).

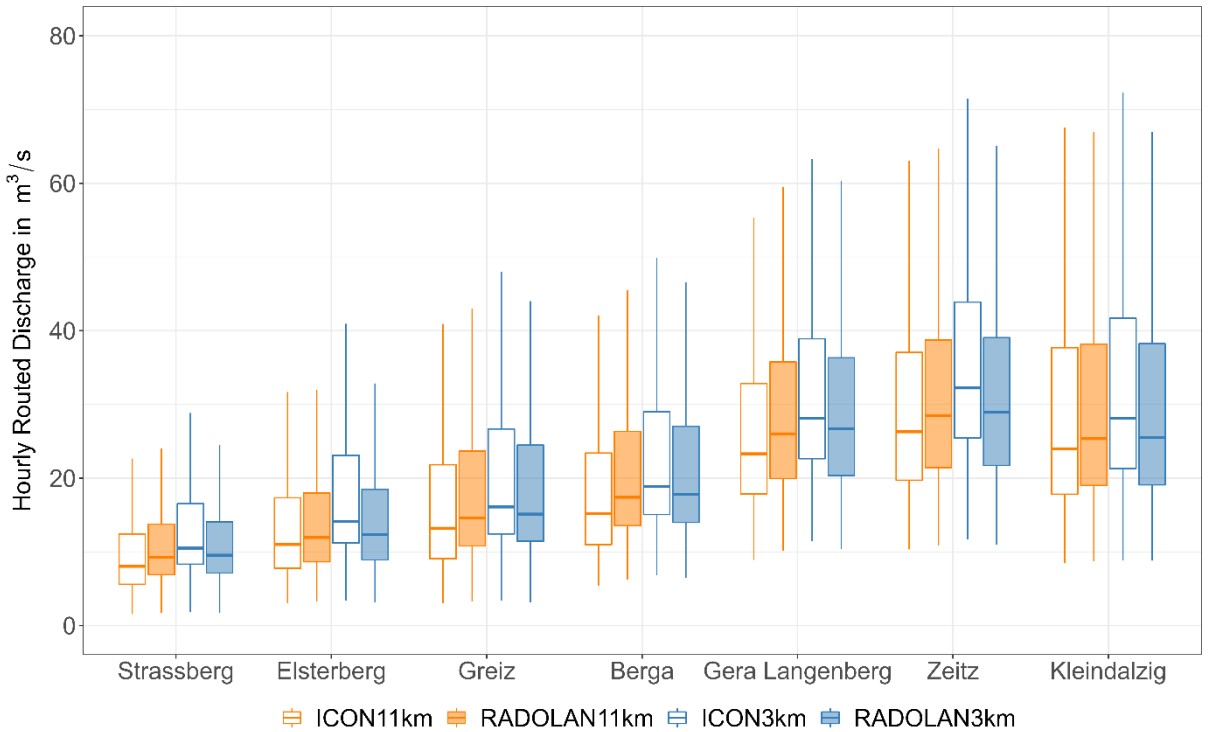

**Fig. 10: Boxplots of hourly routed discharge (period of 2006 to 2014) computed by the WaSiM hydrological model driven with**
**meteorological data from ICON11km and ICON3km, as well as with adjusted radar data of respective equal resolution**





**(RADOLAN11km and RADOLAN3km) for catchments of the main stem of the Weiße Elster river within the study area, showing the median with the 25th and 75th percentile, and whiskers extending to the largest/ smallest value but no more than 1.5-times the interquartile range, while outliers beyond that are not displayed**

The frequency distributions show too frequent low flows for the hydrological model driven with ICON11km and too

frequent high flows when driven with ICON3km for nine out of the twelve studied catchments (with the exception of Eisenhammer, Dröda and Mylau, all of which are independent feeding catchments).

The temporal variability of the monthly mean bias error of hourly routed discharge was found to be strongly shaped by the annual cycle of precipitation bias. During winter (DJF) and early spring (March and April), the hourly discharge estimates based on the climate model data deviate to a similarly strong degree from simulation results obtained based on adjusted radar

data of respective resolution (Fig. 11). However, for summer (JJA) the climate models' biases are markedly different to each other due to the pronounced overestimation of convective precipitation by ICON3km.

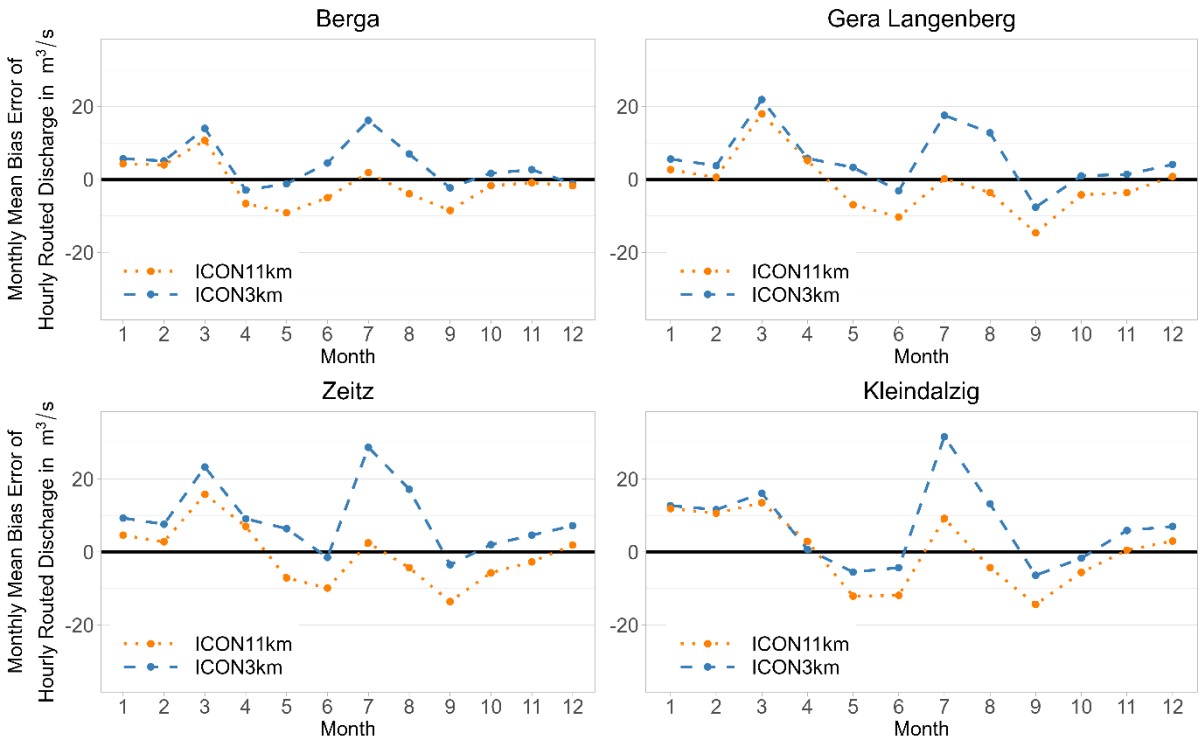

**Fig. 11: Monthly mean bias error of hourly routed discharge from the hydrological model driven with ICON11km/ ICON3km meteorological data compared to when driven with adjusted radar data of equal resolution (RADOLAN11km/ RADOLAN3km)**

**for the four most downstream catchments on the main stem of the Weiße Elster river within the study area over the period of 2006 to 2014**

Driving the hydrological model with the full range of ICON11km meteorological data was found to lead to an underestimation of median hourly routed discharge, but results in an overestimation when looking at the 99.5th percentile individually for six out of the seven catchments on the main stem. The use of ICON3km data as input to the hydrological





model in turn leads to an overestimation of median hourly routed discharge among both the complete dataset and the 99.5th percentile.

### 3.2.3 Event-based Simulation

The model results for the largest flood that occurred during the study period, namely the flood of June 2013, are shown in Fig. 12 exemplarily for the catchment furthest downstream in the study area for which discharge measurements are available.

Both climate models described a heavy precipitation event occurring too shortly with a peak earlier than observed. As a result, the hydrographs simulated based on ICON3km and ICON11km are too narrow and peak too early. The climate models further fell short in depicting antecedent moderate rainfall, which in the hydrological model translates into too low soil moisture, attenuating the flood peaks. It is however noticeable that ICON3km greatly overestimated the maximal daily precipitation, while ICON11km showed an underestimation.





Fig. 12: Top: Daily precipitation estimates over the catchment of Zeitz and those upstream for the period of the 2013 flood for ICON11km, ICON3km, RADOLAN11km and RADOLAN3km, Bottom: the resulting hydrographs (using hourly data) for the catchment of Zeitz together with the discharge measurements




## 4    Discussion

Here we identify strengths and limitations of the studied CPRCM in depicting hourly air temperature, global radiation, relative humidity, wind speed and precipitation over a chosen medium-size river basin in Germany, and compare these to conclusions from other studies. Previous research efforts have mainly focused on precipitation, with only few discussing other meteorological variables (Lucas-Picher et al., 2021) and seldomly for medium to small basins, but rather on larger subnational scale. Furthermore, it is discussed how the performance results from the use of the analysed CPRCM data for

hydrological impact modelling compare to previous studies on other river basins with varying model chains.

### 4.1    Near-Surface Air Temperature

While both the studied RCM (ICON11km) and the CPRCM (ICON3km) were found to compute median hourly air temperatures lower than observed, the deviations fall within the measurement uncertainty bandwidth. Hackenbruch et al. (2016) also showed a cold bias for COSMO-CLM at 2.8 km and 7.0 km resolution over southwestern Germany, and so did

Tölle et al. (2018) for COSMO-CLM at 1.3 km and 7.0 km resolution over central Germany. These studies found higher bias in summer than in winter, however our results show the opposite, likely due to the predominance of cropland and pasture in our study area (c.f. Fig. S1). Over open land, cold biases are amplified due to higher albedo and shorter roughness lengths, the latter favouring the development of stable atmospheric conditions and allowing for stronger nocturnal cooling (Lind et al., 2020). These results underline the importance of scale and exemplarily show how dominating processes on the catchment

scale can differ from those over larger study areas.

The CPRCM was found to offer greatest improvement for air temperature simulations in the summer months, in keeping with a variety of other studies conducted over Europe (Prein et al., 2015). These improvements are likely the result of convection being explicitly resolved leading to better estimates of deep-convective clouds (Leutwyler et al., 2017; Hentgen et al., 2019) and a better representation of convective precipitation (Liu et al., 2017).

### 4.2    Global Radiation

Both ICON3km and ICON11km were found to overestimate median daily global radiation over the study area, however the deviations were not significant under consideration of the measurement uncertainty. Hackenbruch et al. (2016) found an underestimation of half-year mean sums of global radiation over southwestern Germany (1983 – 2000) for COSMO-CLM at 2.8 km and 7.0 km resolution. In our study an underestimation was registered for the highest quantiles of daily global

radiation both in intensity and frequency. ICON3km was able to reduce this bias. Through a better estimation of the onset of convection in the summer mornings and resulting reduced stratification, CPRCMs were shown to be able to dampen the RCMs' positive bias in the estimation of the frequency of high clouds (Keller et al., 2015). The simulation of more frequent clear-sky conditions allows for a reduction in the underestimation of global radiation (Keller et al., 2015; Leutwyler et al., 2017; Hentgen et al., 2019).



### 4.3 Relative Humidity

The skill of CPRCMs in depicting relative humidity is seldomly studied, however it has been shown that an increase in relative humidity constitutes a critical factor in deep convective initiation (Morrison et al., 2022).

No significant differences in mean hourly relative humidity were found for ICON3km or ICON11km compared to the observations under consideration of the measurement uncertainty. Monthly bias means proved negative for winter and autumn, but positive for spring and summer for both the RCM and CPRCM. Hackenbruch et al. (2016) found median relative humidity biases computed by COSMO-CLM at 2.8 km and 7.0 km resolution over southwestern Germany to be positive for both half years, with biases much greater than seen for the ICON-CLM models. Possible explanations for the smaller biases in our study might be improvements to the model physics in the newer ICON-CLM (version 2.6.4) compared to the COSMO-CLM model they used, or differing model parametrisations.

### 4.4 Wind Speed

ICON3km was found to estimate higher wind speed extremes than ICON11km, thereby slightly improving on the underestimation of extremes shown by both climate models. This result is in line with Iles et al. (2020), who found an increase in intensity and frequency of high winds with resolution, when comparing climate models of 12.5 km and 50 km resolution over the European domain. Similar trends were found by Kunz et al. (2010) when comparing REMO-UBA at 10 km resolution and COSMO-CLM at 18 km resolution over Germany. ICON3km improved on the depiction of the frequency of light winds (0.6 to 1.7 m/s) computed by its driving model ICON11km, however both show an overestimation of the frequency of high wind speeds. As such, the simulation of wind speed remains a challenge for both climate models. Belušić et al. (2018) advance that it takes grid spacings of a few kilometres to accurately simulate small-scale wind systems. Hackenbruch et al. (2016) prove added value for the use of a CPRCM of 2.8 km resolution for the computation of channelled wind flow in the Neckar Valley and underline the importance of orographic structures for the initialisation of local wind systems.

### 4.5 Precipitation

ICON11km was found to compute too frequent light precipitation, while underestimating the intensity of heavy rainfall. ICON3km in turn overestimates the intensity of the highest quantiles. These results agree with the consensus from literature (Lucas-Picher et al., 2021), such as with Ban et al. (2021) and Adinolfi et al. (2021). Kendon et al. (2012) showed an RCM of 12 km resolution to compute heavy rainfall as not intense enough, but as too persistent and widespread, in their study over the southern United Kingdom. A CPRCM of 1.5 km resolution in turn achieved improvements in the frequency distribution, duration and spatial extend, but estimated too intense heavy rainfall (Kendon et al., 2012).



An overestimation of the intensity of heavy rainfall by CPRCMs arises as updrafts are computed as too wide under the coarseness of the model grid (Kendon et al., 2023). Additionally, entrainment is insufficiently captured due to turbulence not being sufficiently resolved in the convective cells (Bryan & Morrison, 2012 cited in Quintero et al., 2022).

RCMs with convective parametrisation have shown to underestimate heavy rainfall. In fact, small atmospheric instabilities are sufficient to trigger the convection schemes in the RCMs, leading to the release of light rain and thereby inhibiting a greater computational built-up of the convective cells (Ban et al., 2021). Parametrisation schemes were designed de facto to represent average effects at coarse resolution, however not localised extreme events and individual storms (Kendon et al., 2012).

ICON11km was found to compute too high mean annual precipitation totals, likely because of an overestimation of wet-hour frequency. This result is in agreement with findings by Strandberg & Lind (2020) for a 12.5 km resolution RCM over the European domain. The underestimation of average daily precipitation sums by RADOLAN compared to ground-truth rain gauge data (Kreklow et al., 2020) further exacerbates this positive bias.

The deviations of ICON3km and ICON11km to the adjusted radar data estimates upscaled to their respective resolution were found to be similar between the climate models during autumn and winter but differing most strongly during the peak of summer. In fact, synoptic weather systems, which are dominant in winter, are well represented even in coarse climate models (Strandberg & Lind, 2020) and the CPRCM is likely not providing much additional information or added value to their depiction. For summer however, when small-scale convective processes are at the forefront, the explicit representation of convection in the CPRCM likely leads to pronounced differences compared to estimates by its driving RCM where convection is parametrised (Strandberg & Lind, 2020).

Monthly negative precipitation anomalies were not found to be better represented by ICON3km than by ICON11km. They are predominantly shaped by large-scale atmospheric circulations, handed over to the CPRCM as lateral boundary conditions by its driving RCM. Taylor et al. (2013) however found a change from positive to negative in the sign of the soil-moisture precipitation feedback over their semi-arid domain when switching from an RCM with parametrised convection to a CPRCM. A negative feedback makes a drought less likely, whereas a positive feedback may prolong or intensify it (Taylor et al., 2013). Kendon et al. (2019) further show over an Africa-wide domain, that the use of a CPRCM (4.5 km resolution) can be of added value for the estimation of the length of dry spells through a more accurate depiction of the triggering of diurnal rainfall and an improved representation of the dying out of westward propagating systems.

### 4.6 Hydrological Simulations

All the discussed meteorological variables are used as input to the hydrological model and shape through their biases the accuracy of the discharge simulations. The hydrological model WaSiM driven with ICON3km simulates higher discharge



than when driven with ICON11km. These findings are in agreement with Kay et al. (2015), Mendoza et al. (2016), Reszler et al. (2018), Schaller et al. (2020), Kay (2022), Davis et al. (2022), Ascott et al. (2023) and Poncet et al. (2024).

Given its strong overestimation of precipitation intensity, ICON3km did not allow for improved discharge simulation compared to ICON11km. Both Kay et al. (2015) and Reszler et al. (2018) attributed no added value to the use of their studied CPRCM for hydrological impact modelling. In fact, Kay et al. (2015) found worse performance in discharge computation using the 1.5 km CPRCM compared to using its driving 12 km RCM as input to the gridded hydrological model CLASSIC-GB, given the strong positive precipitation bias of the CPRCM when left uncorrected. Reszler et al. (2018) found deficiencies in the depiction of the temporal distribution of rainfall intensities to be another potential source of error. In a later study, Kay (2022) did however find added value in the use of a 2.2 km CPRCM compared to a 12 km RCM (each 12 member ensembles) for the simulation of daily low flow volume, median flow, and high flow volume across Britain. They attribute these improvements to changes in the physics of the used CPRCM compared to the earlier model employed in their 2015 study (c.f. Kay et al., 2015). Also Mendoza et al. (2016), Schaller et al. (2020), Davis et al. (2022), and Poncet et al. (2024) found various added value in the use of CPRCMs for hydrological modelling (c.f. Table 1).

The 99.5th percentile of hourly discharge was overestimated in the median by the hydrological model driven with ICON3km, as a result of the overestimation of the intensity of heavy precipitation. The input of ICON11km data to the hydrological model also resulted in an overestimation of hourly discharge of the 99.5th percentile in the median, despite an underestimation of precipitation intensity, likely as a result of too frequent light precipitation. Wetter antecedent soil moisture conditions in the climate model with parametrised convection (Hohenegger & Klocke, 2020) are likely to lead to higher runoff and consequently to greater discharge estimates. An overestimation of high discharges by the hydrological model driven with RCM data is also apparent in the study of Reszler et al. (2018). For four out of the six catchments they studied, the return periods from the hydrological model driven with 12.5 km RCM data overtake or at least come to match the return period curves from the observations for high return periods, while undercutting them for lower return periods.

To conclude, no added value could be found in the use of uncorrected climate model data from ICON3km for hydrological impact modelling, neither across its complete range, nor for the 99.5th percentile.

## 5 Conclusion

Climate change projections are key for the development of mitigation and adaptation strategies (IPCC, 2023). Policy decisions rely on climate model results that are not only to be of low bias and uncertainty, but also of high spatial and temporal resolution. CPRCMs bear major potential for advancement towards these requirements, as they run on fine model grids and no longer rely on deep-convection parametrisation schemes, which come with great uncertainty (Ban et al., 2021).

Yet to this day, their performance in depicting meteorological variables on the catchment scale and their potential added value for hydrological impact modelling is scarcely studied (Lucas-Picher et al., 2021).

In the work presented here, the skill of estimating meteorological variables on the catchment scale is analysed for ICON-CLM 2.6.4 in its convection-permitting setup at 3 km resolution (ICON3km), and for its driving model ICON-CLM 2.6.4 at 11 km resolution with parametrised convection (ICON11km, forced with ECMWF-ERA5), exemplarily over the Weiße Elster basin in East Central Germany for the historical period of 2005 to 2014. It is further studied, how well the given uncorrected climate model data is suited for hydrological impact modelling in the catchment over the period of 2006 to 2014

using the distributed physically based hydrological model WaSiM, and whether the CPRCM provides added value.

While ICON3km offers some improvements in the depiction of meteorological variables on the catchment scale, notably for the summer season and in respect to spatial process delineation, however for a variety of other aspects no added value is to be found. ICON3km allows for an improved depiction of wind speed characteristics and shows improvements in the summer-season estimates of mean hourly air temperature and daily global radiation. However, the strong overestimation of

the intensity and frequency of heavy rainfall by ICON3km impedes its suitability for hydrological impact modelling, as it translates into a pronounced overestimation of discharge in the hydrological model. With only a decadal-long time series from one CPRCM over a single basin studied, these results are however only able to give a hint at possible strengths and limitations of CPRCMs. For robust generalised performance indications of CPRCMs, large ensembles and time spans, over varying study areas would be required. Coordinated research efforts are key to achieving this and have started to gain

momentum in recent years (Lucas-Picher et al., 2021). Under consideration of expected model improvements over the next few years (Kendon et al., 2021), the results from this study suggest particular potential of CPRCMs for the simulation of hydroclimatic conditions in summer and for hydrological impact modelling in regions of highly mountainous or urban character.

## 6   Code availability

The code scripts are available from the corresponding author upon request.

## 7   Data availability

Currently the climate model data is only available for partners of the RegIKlim project via the Free Evaluation System Framework (Freva). Open access to the data is planned for the near future.





## 8 Competing interests

The authors declare that they have no conflict of interest.

## 9 Author contribution

OW worked on the methodology, formal analysis, software, visualisation and writing (original draft preparation and editing). VM contributed to the conceptualization, data curation, resources, supervision and review. LB was involved in the funding acquisition, supervision and review.

**10 Acknowledgments**

We thank Klemens Barfus for insightful comments during the development of this paper, as well as Jeewanthi Thotapitiya and Laura Detjen for valuable feedback on the manuscript. Oakley Wagner was financed by the Helmholtz Institute for Climate Service Science (HICSS), a cooperation between Climate Service Center Germany (GERICS) and University of Hamburg, Germany. The climate model data was provided in the frame of the NUKLEUS (Actionable Local Climate
Information for Germany) project, which is part of the larger RegIKlim project funded by the Federal Ministry of Education and Research (BMBF; grant number: 01LR2002A-G). The climate models were run by Klaus Keuler and Michael Woldt at BTU Cottbus-Senftenberg in collaboration with the CLM-Community and EURO-CORDEX. We are grateful for their support. The hydrological results constitute part of the KlimaKonform project at TU Dresden, a sub-project of RegIKlim. Observational meteorological data was provided by Deutscher Wetterdienst (DWD). The catchment-specific setup and
calibration of the hydrological model was built upon an earlier version provided by Sherifdeen Olamilekan Babalola (Babalola, 2023). We thank Deutsches Klimarechenzentrum (DKRZ) and the Centre for Information Services and High-Performance Computing at TU Dresden (ZIH) for generously providing computational resources for this study.



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
