# Peer review of "Do convection-permitting regional climate models have added value for hydroclimatic simulations? A test case over small and mediumsized catchments in Germany"

_EGUsphere, 2025_

## Referee Comment (RC1)

**Review of**

Do convection-permitting regional climate models have added value for hydroclimatic simulations? A test case over small and medium-sized catchments in Germany

By Oakley Wagner, Verena Maleska, and Laurens M. Bouwer

**General Comments**

The presented paper uses the convection permitting regional climate model (CPRCM) ICON-CLM 2.6.4 with the spatial resolution of 3 km together with its driving model with parametrised convection at 11 km resolution for a comparison with measured climate data and as input into a hydrological model in small to medium sized catchments in Eastern Central Germany. Particularly the first part, the comparison with measured climate data, is well and clearly written and provide valuable information about the performance of the models. As the authors stated, it is important to compile more and more example data of CPRCMs to illustrate their possible added value compared to their coarser sister-models (scientific data). This is particularly true regarding the focus on the variables that are crucial for hydrological implications (water budget and floods). Regarding the hydrological modelling part, however, I doubt that the corresponding chapters are ready for publication without revisions (see specific comments). In its current state, the modelling work does not provide additional findings about the usage of these RCMs for climate impact studies. In particular, there are no conclusions given regarding the impact of the different model input variables on the model output, in this case also the representation of floods. Of course, the huge overestimation of precipitation intensities in the CPRCM will play the main role. In this respect, I miss conclusions about the impact of such an overestimation on the flood representation, particularly the non-linear behaviour of the flood generation (threshold processes) compared to the calibrated case and gauge data. This also implies a more comprehensive evaluation of the used hydrological model to accurately represent these processes within the calibration and validation procedure.

**Specific comments**

P. 6 / L. 88: Are the stations used here implemented in the RADOLAN scheme? If so, are the station values preserved after regionalisation in RADOLAN? Please give a short clarification.

P. 6 / L. 102: Are the climate models run on hourly time step? Is this the effective temporal resolution? Please add a few words or point to the reference.

P 7 / L. 122: Please give the reason to choose the Sturges' rule.

P 7 / L. 136: How do the THIESSEN interpolated rainfall compare to the RADOLAN product? See first comment above. Please add a short comment.

P. 8 / L. 145: The bandwidth is hard to read. I assume it is ±0.08 to ±0.76 K. Please clarify.

P. 8 / L. 154 and further lines and plots (e.g., Fig. 2) in the manuscript: please remove "error" in "monthly mean bias error", this is redundant and may mislead.

P. 10 / Fig. 3: For clarity, I would recommend to make 4 plots for the 4 seasons out of this plot.

P. 11 / L. 200ff: Please rewrite the sentence to clarify and add the general measurement uncertainties (or add references).

P 12 / after Fig. 5: Please add a short chapter of the calculated potential evapotranspiration (I assume, ET0 by the PENMAN-MONTEITH formula), since this is a relevant input variable and includes

the mentioned variables. Please also give a short comment about the partial influence of the variables (probably add the ET0 formula). Systematic errors may be relevant in the context of error propagation through the model chain, particularly regarding water budget and soil moisture.

P. 13 / L. 225: Please discuss shortly this (huge) overestimation of the 3km model. Are there many hourly intensities in the 3km model in the range of 100 to 150 mm/hour? Are these single intensities or embedded into longer rainfall events? What does this mean for flood representation in small catchments (flash floods)?

P. 14 / Fig. 7: Please give the threshold for the 99.5 % percentile (app. 2 mm/hour?).

P 15 / L. 246 ff: Is there a flood season? If so, please add the information, in which season mainly the (large) floods occur (seasonality of floods)? Do snow melt induced floods play a role? If so, please refer to the temperature data evaluation.

P. 16 / Fig. 9: Is there a reason to use this kind of presentation (anomalies)? In my view, the monthly values can also be explicative.

P. 16 / L. 269 fff: Please add which precipitation input data where used for calibration, the THIESSEN interpolated or the RADOLAN? How were the hydrological model parameters chosen, which were fixed, and what were the main parameters calibrated? Please add the graphs of the two events for the catchments and quote the (estimated) return periods? How is the performance at smaller events? Is there a non-linear shift in runoff generation from smaller to larger events? Can this be identified in the measured gauge data?

P. 17 / L. 280: In my opinion, this assumption can only be made, if the model adequately represents the main runoff generation processes (water budget, flood generation - non-linearity) and error propagation can be quantified. Therefore, please show the calibration and validation results in a more comprehensive way (see above).

P. 17 / Chapter 3.2.2: The model was calibrated to one single event and validated to another single event. Please show, that the model accurately captures the long term water balance. Please add the observation (or the calibration - is RADOLAN 3km closest to the calibration?) results into Fig. 10 and discuss the differences. Particularly in the context of long term model runs, in a climate change framework occurring systematic errors (uncorrected precipitation and potential evaporation) may lead to an accumulation of errors. In this respect, please discuss also possible biases in the ET0 and the impact on soil moisture simulation. Please also show what flood event peaks are "produced" by the extraordinary high rainfall intensities. Probably give short flood peak statistics (annual maxima or POT - peaks over a certain threshold).

P. 17 / Fig. 10 and further lines and plots: please remove "routed", just use "discharge".

P. 18 / Fig. 11: Please add the catchment sizes of the four examples. Also, the relative differences (in %) would be good to mention and to compare to the corresponding (relative) differences in the catchment precipitation.

P. 18 / L. 307: Please clarify "full range of ICON11km meteorological data" or rewrite.

P. 19 / L. 13: Please repeat for clarification that this is the calibration event (e.g., in brackets).

P. 19 & 20 / Fig. 12: I believe this chart requires further discussion. Again: Are the RADOLAN 3km and 11km driven simulations close to the calibration and, can they be seen as reference simulation? Please add a short comment. If so, it seems to me, that the calibration focused on the representation of the flood peak. In my view, with the corresponding graphs, this could be discussed in detail in

chapter 3.2.1. Also, the performance regarding the annual and seasonal water balance (discharge volumes) together with the soil moisture and storage simulation should be analysed. The smaller preceding events are significantly overestimated with the RADOLAN input, so one can assume, that the antecedent conditions at the start of the main event are overestimated as well. In this respect, it should be analysed, if the different event sizes and event types are generally represented well by the calibrated model(s) (frequency).

Please add, how the initial conditions in May 2013 were chosen. Were they obtained by the continuous simulation? The high discharge in the RADOLAN driven simulation implies also high soil moisture and storage fillings in the simulation and furthermore, lower antecedent losses and a higher initial discharge at the main event. The observed discharge indicates, that the smaller rainfall events recorded in the RADOLAN data did not lead to a discharge rise, which could be related to a lower soil moisture status in May than simulated.

With the climate model input the simulation results preceding the event are closer to the observation. It would be interesting, how the results would look like if the same antecedent conditions were used for all model inputs. This would help to clarify the impact of the different rainfall input at the particular event and also the role of the initial catchment conditions, i.e. soil moisture. This would lead to the question, if the systematic errors in precipitation and possibly also in ET0, may be more relevant and distort the interpretation of single flood events.

Generally, the underestimation of the flood by the ICON 3km driven model is a rather surprising result, because this model setup considerably overestimates the discharge in the mean and in all presented percentiles in all catchments (Fig. 10). The initial baseflow-to-peak rise, however, seems to be similar to the RADOLAN driven models (or even larger). In any case, these discrepancies have - in my view - to be further addressed.

P. 21 / Chpt. 4: In my view, the discussion needs revision after further analyses have been performed.

P. 21f / Chpt. 4.1, 4.2, 4.3, and 4.4: Please add the implications on hydrology, e.g., snow melt, ET (water budget).

P. 22 / Chpt. 4.5: Please add a paragraph about the impact of the huge precipitation overestimation (refer, e.g., to depth-duration frequency curve (Fig. 8)) on the hydrological model output (non-linearity). What does this mean for flood peaks in terms of shifting of return periods (please also refer to literature). What are the implications for future predictions when such biases occur?

P. 22 / L: 379: ICON3km overestimates the intensity of the highest quantiles. Apparently, it fails at the event. Can this be explained? See above.

P. 23 / Chpt. 4.6: Hydrological simulations: it would be interesting, what event peaks were simulated with these extraordinary intensities (higher than 2013, see Fig. 6). Please provide some conclusions regarding flood statistics. In general, please discuss shortly the usability or advantages of the application of the complex model with lots of input variables and model parameters that are usually difficult to measure (e.g. soil hydraulic conductivity) as well as the large computation time (numerical solution of Richard's Equation), when such high uncertainties in precipitation occur. Can it be recommended for larger catchments, sensitivity analyses or ensemble modelling (e.g., sensitivity of the hydraulic conductivities)?

P. 24 / Chpt. 5 Conclusion: Would you recommend to perform more local studies or to set the focus on larger scale studies? The point-by-point comparison of the meteorological variables, which is a rather strict test, can also be carried out on larger scale. Please add a concluding comment about the

applicability of the model chain for future predictions, when such high precipitation biases occur. Would bias correction makes sense?

---

## Author Comment (AC1)

**Review of**

Do convection-permitting regional climate models have added value for hydroclimatic simulations? A test case over small and medium-sized catchments in Germany

By Oakley Wagner, Verena Maleska, and Laurens M. Bouwer

**General Comments**

The presented paper uses the convection permitting regional climate model (CPRCM) ICON-CLM 2.6.4 with the spatial resolution of 3 km together with its driving model with parametrised convection at 11 km resolution for a comparison with measured climate data and as input into a hydrological model in small to medium sized catchments in Eastern Central Germany. Particularly the first part, the comparison with measured climate data, is well and clearly written and provide valuable information about the performance of the models. As the authors stated, it is important to compile more and more example data of CPRCMs to illustrate their possible added value compared to their coarser sister-models (scientific data). This is particularly true regarding the focus on the variables that are crucial for hydrological implications (water budget and floods). Regarding the hydrological modelling part, however, I doubt that the corresponding chapters are ready for publication without revisions (see specific comments). In its current state, the modelling work does not provide additional findings about the usage of these RCMs for climate impact studies. In particular, there are no conclusions given regarding the impact of the different model input variables on the model output, in this case also the representation of floods. Of course, the huge overestimation of precipitation intensities in the CPRCM will play the main role. In this respect, I miss conclusions about the impact of such an overestimation on the flood representation, particularly the non-linear behaviour of the flood generation (threshold processes) compared to the calibrated case and gauge data.

We thank the reviewer for these comments, for identifying this gap and for the interesting direction of resolution effects on the representation of non-linear behaviour. We have done several additional analyses in response, and propose to make several adjustments to the paper, and give more explanations and answers below. We hope that these answers and adjustments will help to improve our paper, and would make it acceptable for publication.

Disentangling the effects of bias of individual meteorological variables on the flood generation is particularly challenging since many of the key processes in hydrological modelling are affected by the whole range of meteorological input variables. We want to give this more thorough consideration, on the one hand, by giving an overview of the input requirements of the individual modules of WaSiM and, on the other hand, by expanding our focus beyond discharge, conducting analyses on evapotranspiration and soil moisture. A look into threshold processes will allow to merge and extend the gained insights.

The modules in WaSiM for high-resolution hydrological simulation in the temperate zone are for processes of evapotranspiration, snow accumulation and snow melt, interception of snow and precipitation, as well as an unsatured-zone model and a groundwater model. Potential evapotranspiration is calculated after Penman-Monteith, thereby showing dependency on the meteorological input variables of air temperature, global radiation, relative humidity and wind speed. Interception processes are described by an energy balance approach, considering the complete suite of meteorological input variables, just as is the case for the simulation of snow accumulation and snow melt. The unsaturated zone model is built on the Richards-equation and draws on the aforementioned modules, as well as on the groundwater model.

The soil moisture bias in the hydrological model driven with ICON11km is found to be negative compared to when driven with RADOLAN11km (see Fig. 1 below), which is likely a reflection of the underestimation of precipitation intensity by ICON11km. Looking at the model driven with ICON3km, the sign of the bias changes for moderate to high soil moisture estimates given the relative overestimation of rainfall intensities by ICON3km compared to RADOLAN3km. We will include this new QQ-plot in the revised paper. Higher soil moisture in the hydrological model driven by ICON3km is likely to lead to higher runoff in case of a moderate to heavy rainfall event. Soil moisture is further shaped by total evapotranspiration but becomes the limiting factor for the latter in summer.

Fig. 1: QQ-plots for relative hourly soil moisture, as calculated by the hydrological model driven with ICON11km and RADOLAN11km, respectively ICON3km and RADOLAN3km for the period 2006 to 2014

In fact, a change in sign of bias is also visible for evapotranspiration estimates in the summer months, when comparing ICON11km and ICON3km (see Fig. 2 below). Besides higher water availability, a reduction of the negative summer bias in temperature and global radiation by ICON3km may contribute to the (slightly) higher ET estimates.

Fig. 2: Monthly mean bias of hourly total evapotranspiration, as calculated by the hydrological model driven with ICON3km and ICON11km for the period 2006 to 2014

We further looked into threshold processes by analysing the hydrological response of runoff volume ( $Q_{tot}$ ) to the sum of rainfall volume ( $T_{tot}$ ) and antecedent rainfall (AT). The choice of considered meteorological factors was based on Ross et al. (2021), who found non-linear behaviour in the flood generation to be most commonly present when using the  $T_{tot}$  + AR threshold. We employed a modified version of the HydRun tool (Tang & Carey, 2017; with modifications to the runoff peak search algorithm) for the rainfall-runoff-attribution and chose an antecedent rainfall window of 8 hours. We discarded the months of November to May to avoid the effects of snow melt. Fig. 3 shows the relation between runoff and the sum of total rainfall volume and antecedent rainfall for simulations with ICON11km, ICON3km, RADOLAN11km and RADOLAN3km. The segmented regression is highly sensitive to rainfall-runoff events of high return period, such as predominantly recorded from the ICON model simulations. As such, no firm conclusions can be drawn from Fig. 3 on the existence or position of a break point in the relation between runoff and rainfall.

We will include these evaluations in the revised paper.

Fig. 3: Upper half: Scatter plots of total runoff ( $Q_{tot}$ , min. peak threshold for runoff events: 0.03 mm/h) to the sum of total rainfall ( $R_{tot}$ ) and 8-h antecedent rainfall (AR) over the catchment of Kleindalzig and those upstream, fitted with a piecewise linear regression model. The thresholds are indicated with a dotted vertical line. Lower half: Combined piecewise linear regression model results of  $Q_{tot}$  to  $R_{tot}$  + AR

This also implies a more comprehensive evaluation of the used hydrological model to accurately represent these processes within the calibration and validation procedure.

We calibrated the model on the flood of June 2013, which is the largest flood observed in the catchment during the study period. At Zeitz, peak discharges were measured of an estimated return period of 100 years (LHW, 2014). The model was further calibrated for its long-term statistics on the discharge measurements of the year 2012. The year of 2012 was chosen as it shows comparatively low monthly precipitation anomalies. We looked at weekly discharge sums, whereby the weeks of the year were defined according to ISO 8601.

The second largest flood, which occurred in January 2011 as a result of snowmelt, was used for validation despite having a different genesis, since the short study period did not capture another significant summer flood. The return period of the measured peak discharge at the gauge Zeitz was of 10 years and at the gauge Kleindalzig of > 25 years (LHW, 2011). The model was further validated on the calendar year of 2008, another year of low monthly precipitation anomalies.

The calibration results on the summer flood of 2013 (see Fig. 4 below) are considered satisfactory, except for the headwater catchment of Weida. However, the model did not capture the steep peak (flashiness) of the rainfall-runoff response and the responsiveness of the baseflow. The validation results underline these conclusions and show an overestimation of direct discharge from snowmelt, as well as too high and retarded flood peaks. We will add this explanation in the revision of the paper.

Looking at a whole calendar year, the weekly discharge estimates show a positive bias in the direct discharge, interflow and baseflow simulations.

**Specific comments**

P. 6 / L. 88: Are the stations used here implemented in the RADOLAN scheme? If so, are the station values preserved after regionalisation in RADOLAN? Please give a short clarification.

We used the processed RADOLAN data set offered via the ReKIS portal, which covers the German federal states of Saxony, Saxony-Anhalt and Thuringia (Körner, 2022). The native RADOLAN product builds the foundation but has further been gap-filled using the ground stations of the German Weather Service (DWD). Data from DWD's precipitation gauges is freely available and has been used for calibrating and validating our hydrological model. The same station network was employed by DWD in the adjustment of the radar measurements in the RADOLAN scheme (Winterrath et al., 2012). The choice of adjustment procedure is determined based on a performance test with a few control stations (see Winterrath et al., 2012). It is unknown whether any, respectively which of these control stations correspond precisely to the stations used in our study, but in principle the station values should be preserved. We will clarify this in the revision of the paper.

P. 6 / L. 102: Are the climate models run on hourly time step? Is this the effective temporal resolution? Please add a few words or point to the reference.

The analysed climate model data was outputted by the ICON-CLM model in an hourly resolution and provided to users in the frame of the NUKLEUS (Actionable Local Climate Information for Germany) project.

P 7 / L. 122: Please give the reason to choose the Sturges' rule.

Sturges' rule has established itself as the first choice in finding the optimal number of bins in a histogram (Scott, 2009). It should however indeed be noted that for large datasets it is prone to oversmoothing (Scott, 2009).

P 7 / L. 136: How do the THIESSEN interpolated rainfall compare to the RADOLAN product? See first comment above. Please add a short comment.

Fig. 5 shows the QQ-plots for catchment spatial averages of the Thiessen-interpolated hourly rainfall estimates from RADOLAN3km, resp. RADOLAN11km to the catchment spatial averages of the Thiessen-interpolated hourly rain gauge measurements, as computed in a preprocessing step by the hydrological model WaSiM.

Fig. 5: QQ-plots for catchment spatial averages of the Thiessen-interpolated hourly rainfall estimates from RADOLAN3km, resp. RADOLAN11km to the catchment spatial averages of the Thiessen-interpolated hourly rain gauge measurements for the period 2005 to 2014

The Thiessen-interpolated RADOLAN products derive most strongly from the Thiessen-interpolated station measurements for moderate to high intensity rainfalls. We looked exemplarily at the effect this has on the simulation of the largest flood in the catchment during the study period (Fig. 6 below) by adding the calibration results into Fig. 12 of the original submission. The deviations of the discharge simulations driven with RADOLAN data from those driven with precipitation station data are found to be minor. We will include these explanations in the revised paper and add the corresponding figures to the appendix.

Fig. 6: Daily spatial average precipitation estimates over the catchment of Zeitz and those upstream for the period of the 2013 flood for ICON11km, ICON3km, RADOLAN11km, RADOLAN3km and the interpolated precipitation station measurements, Bottom: the resulting hydrographs (using hourly data) for the catchment of Zeitz together with the discharge measurements

P. 8 / L. 145: The bandwidth is hard to read. I assume it is ±0.08 to ±0.76 K. Please clarify.

Yes, that's right. We will clarify this in the revised paper.

P. 8 / L. 154 and further lines and plots (e.g., Fig. 2) in the manuscript: please remove "error" in "monthly mean bias error", this is redundant and may mislead.

Thank you for pointing this out. We will change it.

P. 10 / Fig. 3: For clarity, I would recommend to make 4 plots for the 4 seasons out of this plot.

Thank you. We will follow the suggestion.

P. 11 / L. 200ff: Please rewrite the sentence to clarify and add the general measurement uncertainties (or add references).

The German Weather Service commonly measures wind speed with the Ultrasonic Anemometer 2D, which has an accuracy of  $\pm$  0.1 m/s for wind speed below 5 m/s and of  $\pm$  1.5% for wind speeds higher than that (METEK, n.d.). We will clarify this in the manuscript.

P 12 / after Fig. 5: Please add a short chapter of the calculated potential evapotranspiration (I assume, ETO by the PENMAN-MONTEITH formula), since this is a relevant input variable and includes the mentioned variables. Please also give a short comment about the partial influence of the variables (probably add the ETO formula). Systematic errors may be relevant in the context of error propagation through the model chain, particularly regarding water budget and soil moisture.

We will clarify that potential evapotranspiration was calculated by the Penman-Monteith formula. This approach has established itself as the standard in hydrological modelling (Ndulue & Ranjan, 2021). A range of studies have analysed how uncertainties in meteorological variables shape the estimation of current and future potential evaporation and evapotranspiration using the Penman-Monteith method (e.g. Meyer et al., 1989; Kay & Davies, 2008; Lai et al., 2022). We will include this information in the revised paper. It shall be noted that while evapotranspiration greatly influences the long-term water balance, however for heavy rainfall events, as are the focus of this paper, evapotranspiration is of secondary importance given that soil saturation is quickly reached.

P. 13 / L. 225: Please discuss shortly this (huge) overestimation of the 3km model. Are there many hourly intensities in the 3km model in the range of 100 to 150 mm/hour? Are these single intensities or embedded into longer rainfall events? What does this mean for flood representation in small catchments (flash floods)?

We will take a closer look at the individual rainfall events and their associated runoff events, as identified using the HydRun tool (Tang & Carey, 2017). A particular focus will be placed on extreme intensity rainfall estimates, their occurrence and effects on runoff generation.

P. 14 / Fig. 7: Please give the threshold for the 99.5 % percentile (app. 2 mm/hour?).

The 99.5% percentile of hourly precipitation by RADOLAN3km is at 2.8 mm/h and by RADOLAN11km at 2.7 mm/h.

P 15 / L. 246 ff: Is there a flood season? If so, please add the information, in which season mainly the (large) floods occur (seasonality of floods)? Do snow melt induced floods play a role? If so, please refer to the temperature data evaluation.

In the study region, floods occur primarily in summer as a result of high-intensity convective storms, as well as in winter due to prolonged rainfall, sometimes amplified by rapid snowmelt. CPRCMs are expected to offer little improvement for the representation of winter floods, as these are governed by large-scale synoptic weather systems (Strandberg & Lind, 2021), but CPRCMs have great potential for the simulation of local convective summer storms. The focus of our paper is therefore on summer convective rainfall events.

**P. 16 / Fig. 9: Is there a reason to use this kind of presentation (anomalies)? In my view, the monthly values can also be explicative.**

We looked at the precipitation anomalies to the long-term mean to differentiate whether the climate models are able to depict months with (1) unusual wet conditions and (2) unusual dry conditions. We identified a difference in performance for the two cases, with ICON3km overestimating the first, but in contrast proving able to capture even deep negative precipitation anomalies.

For reference, the monthly precipitation sums averaged over the study area are shown in Fig. 7 for the ICON models and the RADOLAN data. In contrast, the precipitation anomalies of the ICON models (Fig. 9 in the paper) were computed only for cells overlaying rain gauges and expressed in relation to the long-term measurements of the latter.

Fig. 7: Monthly precipitation sums averaged over the study area for ICON11km, ICON3km, RADOLAN11km, RADOLAN3km for the period of 2005 to 2014

P. 16 / L. 269 fff: Please add which precipitation input data where used for calibration, the THIESSEN interpolated or the RADOLAN? How were the hydrological model parameters chosen, which were fixed, and what were the main parameters calibrated? Please add the graphs of the two events for the catchments and quote the (estimated) return periods? How is the performance at smaller events? Is there a non-linear shift in runoff generation from smaller to larger events? Can this be identified in the measured gauge data?

P. 17 / L. 280: In my opinion, this assumption can only be made, if the model adequately represents the main runoff generation processes (water budget, flood generation - non-linearity)

and error propagation can be quantified. Therefore, please show the calibration and validation results in a more comprehensive way (see above).

We used the distributed physically-based hydrological model WaSiM, which we ran with Thiessen-interpolated meteorological station data for calibration. Four main parameters were used for calibration, namely (1) the single reservoir recession constant for surface runoff, (2) the single reservoir recession constant for interflow, (3) the drainage density for interflow and (4) the fraction of surface runoff on snow melt. More information on the hydrological model is given by Schulla (2021).

A discussion of the calibration and validation results is given as an answer to the general comments.

Fig. 8 shows a scatter plot of total runoff ( $Q_{tot}$ , estimated from the measured discharge) to the sum of thiessen-interpolated measured total rainfall ( $R_{tot}$ ) and 8-h antecedent rainfall (AR) over the catchment of Zeitz and those upstream. The data is widely scattered and doesn't allow to draw conclusions on a non-linear behaviour of flood generation.

Fig. 8: Scatter plot of total runoff ( $Q_{tot}$ , estimated from the measured discharge with a peak threshold for runoff events of 0.015 mm/h) to the sum of thiessen-interpolated measured total rainfall ( $R_{tot}$ ) and 8-h antecedent rainfall (AR) over the catchment of Zeitz and those upstream

P. 17 / Chapter 3.2.2: The model was calibrated to one single event and validated to another single event. Please show, that the model accurately captures the long term water balance. Please add the observation (or the calibration - is RADOLAN 3km closest to the calibration?) results into Fig. 10 and discuss the differences. Particularly in the context of long term model runs, in a climate change framework occurring systematic errors (uncorrected precipitation and potential evaporation) may

lead to an accumulation of errors. In this respect, please discuss also possible biases in the ETO and the impact on soil moisture simulation. Please also show what flood event peaks are "produced" by the extraordinary high rainfall intensities. Probably give short flood peak statistics (annual maxima or POT - peaks over a certain threshold).

- Regarding the calibration and validation results, please refer to the answers given to the general comments above.
- Regarding the comparison of the discharge simulations by the hydrological model driven with RADOLAN3km and the interpolated rainfall gauge measurements, please refer to the answers given to the comment to P 7 / L. 136.
- Regarding the biases in ETO and the impact on soil moisture simulation, please refer to the answers given to the general comments.
- We will conduct analyses on the simulated extreme intensity rainfall estimates, looking at their occurrence and effects on runoff generation under application of flood peak statistics.

P. 17 / Fig. 10 and further lines and plots: please remove "routed", just use "discharge".

Thank you for the comment. We will implement it.

P. 18 / Fig. 11: Please add the catchment sizes of the four examples. Also, the relative differences (in %) would be good to mention and to compare to the corresponding (relative) differences in the catchment precipitation.

The catchment of Berga has an area of 195 km², Gera Langenberg of 303 km², Zeitz of 295 km² and Kleindalzig of 425 km². We will clarify this in the manuscript.

We computed the mean monthly sums of catchment precipitation and discharge estimates, and deduced the relative biases of ICON3km and ICON11km to RADOLAN3km and RADOLAN11km respectively (Fig. 9 below). In the paper we show how both ICON3km and ICON11km overestimate mean yearly precipitation sums. These findings are in line with the results of the mean monthly sums shown below. Only during the peak of summer do ICON3km and ICON11km show biases of markedly different extent, with higher positive biases by the CPRCM (ICON3km). The overestimation of summer precipitation translates into the discharge simulations. A direct comparison of the relative biases is hindered by the difference in reference and scale of the absolute values from which they have been derived.

Fig. 9: Relative deviations of average monthly sums of catchment precipitation and discharge estimates for ICON3km and ICON11km to RADOLAN3km and RADOLAN11km for the four most downstream catchments on the main stem of the Weiße Elster river within the study area over the period of 2006 to 2014

P. 18 / L. 307: Please clarify "full range of ICON11km meteorological data" or rewrite.

We will rewrite this sentence to: "Driving the hydrological model with ICON11km meteorological data was found to lead to an underestimation of median hourly discharge, but results in an overestimation when looking only at the 99.5th percentile of hourly discharge for six out of the seven catchments on the main stem."

P. 19 / L. 13: Please repeat for clarification that this is the calibration event (e.g., in brackets).

We will add this clarification in the manuscript.

P. 19 & 20 / Fig. 12: I believe this chart requires further discussion. Again: Are the RADOLAN 3km and 11km driven simulations close to the calibration and, can they be seen as reference simulation? Please add a short comment.

This has been discussed previously as an answer to the comment to P 7 / L. 136.

If so, it seems to me, that the calibration focused on the representation of the flood peak. In my view, with the corresponding graphs, this could be discussed in detail in chapter 3.2.1. Also, the performance regarding the annual and seasonal water balance (discharge volumes) together with the soil moisture and storage simulation should be analysed.

Soil moisture has been studied and discussed as a response to the general comments.

The smaller preceding events are significantly overestimated with the RADOLAN input, so one can assume, that the antecedent conditions at the start of the main event are overestimated as well.

In this respect, it should be analysed, if the different event sizes and event types are generally represented well by the calibrated model(s) (frequency).

Please add, how the initial conditions in May 2013 were chosen. Were they obtained by the continuous simulation? The high discharge in the RADOLAN driven simulation implies also high soil moisture and storage fillings in the simulation and furthermore, lower antecedent losses and a higher initial discharge at the main event. The observed discharge indicates, that the smaller rainfall events recorded in the RADOLAN data did not lead to a discharge rise, which could be related to a lower soil moisture status in May than simulated. With the climate model input the simulation results preceding the event are closer to the observation. It would be interesting, how the results would look like if the same antecedent conditions were used for all model inputs. This would help to clarify the impact of the different rainfall input at the particular event and also the role of the initial catchment conditions, i.e. soil moisture. This would lead to the question, if the systematic errors in precipitation and possibly also in ETO, may be more relevant and distort the interpretation of single flood events.

Generally, the underestimation of the flood by the ICON 3km driven model is a rather surprising result, because this model setup considerably overestimates the discharge in the mean and in all presented percentiles in all catchments (Fig. 10). The initial baseflow-to-peak rise, however, seems to be similar to the RADOLAN driven models (or even larger). In any case, these discrepancies have – in my view - to be further addressed.

Fig. 10 expands the hyetograph-hydrograph-plot of the 2013 flood (Fig. 12 in the manuscript) by the soil moisture time series, as gained from continuous hydrological simulation. ICON11km climate input data leads to too low soil moisture, a finding in keeping with Fig. 1 of this document. As a consequence, the flood wave is greatly buffered by the soils, leading to a rise in soil moisture and a comparatively low flood peak. Driven with RADOLAN3km and RADOLAN11km, the soil moisture results suggest that the storm rained off over saturated soils; the peak rainfall does not lead to an increase of soil moisture, but to a pronounced runoff peak. While on average, soils in the hydrological model driven with ICON3km showed similar initial soil moisture values, however the rainfall likely fell over catchments with not yet fully exhausted storage capacities. As a consequence, average soil moisture rose further and the flood peak was attenuated. ICON3km was run over the vast Central European (CEU) domain, forced three-hourly at its boundaries by ECMWF-ERA5 reanalysis data. While the general events and patterns match, the climate model, as is widely known, is unlikely to perfectly match the actual geographic position of a convective storm and spatial offsets are to be expected.

We will replace Fig. 12 in the manuscript by the extended version including the soil moisture time series (Fig. 10 below) and discuss the results.

Fig. 10: Top: Daily spatial average precipitation estimates over the catchment of Zeitz and those upstream for the period of the 2013 flood for ICON11km, ICON3km, RADOLAN11km and RADOLAN3km, Centre: the respective hourly relative soil moisture spatial averages, Bottom: the resulting hydrographs (using hourly data) for the catchment of Zeitz together with the discharge measurements

P. 21 / Chpt. 4: In my view, the discussion needs revision after further analyses have been performed.

We will revise and extend the discussion with the new analyses.

P. 21f / Chpt. 4.1, 4.2, 4.3, and 4.4: Please add the implications on hydrology, e.g., snow melt, ET (water budget).

In the revised manuscript, we will discuss the results on the implications on hydrology presented above as a reply to the general comments.

P. 22 / Chpt. 4.5: Please add a paragraph about the impact of the huge precipitation overestimation (refer, e.g., to depth-duration frequency curve (Fig. 8)) on the hydrological model output (non-linearity). What does this mean for flood peaks in terms of shifting of return periods (please also refer to literature). What are the implications for future predictions when such biases occur?

In the planned analyses of individual rainfall events and their associated runoff events, we will also look into flood statistics and potential shifts in return periods by varying modelling resolution. However, the short time series of 9 years only captured a limited number of floods, posing a strong limitation to this endeavour.

P. 22 / L: 379: ICON3km overestimates the intensity of the highest quantiles. Apparently, it fails at the event. Can this be explained? See above.

In the case study of the 2013 flood event, ICON3km showed an overestimation of peak rainfall intensity, but the flood peak was underestimated. As outlined as a response to the comment to P. 19 & 20 / Fig. 12, the simulated rainfall cell was likely shifted in space and rained off over soils which were not yet full saturated, while in the RADOLAN3km-driven simulation, the rainfall hit saturated soils and was converted integrally into runoff.

P. 23 / Chpt. 4.6: Hydrological simulations: it would be interesting, what event peaks were simulated with these extraordinary intensities (higher than 2013, see Fig. 6). Please provide some conclusions regarding flood statistics.

Thank you for pointing us in this direction. We will add some comments in the revised paper on the flood statistics, under consideration of the literature, given that the time series used here is too short to estimate high return periods.

In general, please discuss shortly the usability or advantages of the application of the complex model with lots of input variables and model parameters that are usually difficult to measure (e.g. soil hydraulic conductivity) as well as the large computation time (numerical solution of Richard's Equation), when such high uncertainties in precipitation occur. Can it be recommended for larger catchments, sensitivity analyses or ensemble modelling (e.g., sensitivity of the hydraulic conductivities)?

The added value of process-based fully distributed hydrological models compared to conceptual lumped models for discharge simulations driven with convection-permitting regional climate models is a field of ongoing research. Poncet et al. (2024) used hourly convection-permitting regional climate model data to drive the process-based distributed hydrological model CREST and the conceptual lumped model GR5H. They found an

improvement in the simulation of flood intensity and frequency using the CPRCM compared to a RCM. They further studied the number of peaks over threshold and found CREST to overestimate, while GR5H offered better agreement with observations. These results show that conceptual lumped models remain valid, but more research is needed to draw firm conclusion, since deterministically better process representation, as in process-based models, goes in hand with reduced bias. A major bottleneck of process-based fully distributed models is however their high computational cost. We will add some of these points in the revision.

P. 24 / Chpt. 5 Conclusion: Would you recommend to perform more local studies or to set the focus on larger scale studies? The point-by-point comparison of the meteorological variables, which is a rather strict test, can also be carried out on larger scale.

Climate model data of high spatial and temporal resolution is needed for local hydrological impact studies in small catchments of mountainous or highly urban character (e.g. Schaller et al., 2020; Tamm et al., 2023). Here CPRCMs have the potential to provide a real improvement. It is therefore recommended to direct research to complex study sites where coarser RCMs with parametrised convection fall short. We will add text along these lines to the paper.

Please add a concluding comment about the applicability of the model chain for future predictions, when such high precipitation biases occur. Would bias correction makes sense?

High precipitation bias poses a strong limitation to the applicability of the model chain, however bias correction of climate model data of high spatial and temporal resolution, as needed for hydrological impact modelling, comes with major challenges (e.g. Haerter et al., 2011; Addor & Seibert, 2014). Bias correction furthermore relies on the assumption of stationary bias, making it questionable for climate projections (Huang, Krysanova & Hattermann, 2014). A correction of the climate model data is outside the scope of our paper.

Given that a number of studies have indeed identified added value in the use of CPRCMs (e.g. Rudd et al., 2020; Kay et al., 2022; Poncet et al. 2024), an ensemble approach would be recommended. Coordinated research efforts are needed and have gained momentum in recent years (Lucas-Picher et al., 2021). We will add a concluding remark on this.

**References**

- Addor, N. and Seibert, J.: Bias correction for hydrological impact studies beyond the daily perspective, Hydrol. Process., 28, 4823–4828, https://doi.org/10.1002/hyp.10238, 2014.
- Haerter, J. O., Hagemann, S., Moseley, C., and Piani, C.: Climate model bias correction and the role of timescales, Hydrol. Earth Syst. Sci., 15, 1065–1079, https://doi.org/10.5194/hess-15-1065-2011, 2011.
- Huang, S., Krysanova, V., and Hattermann, F. F.: Does bias correction increase reliability of flood projections under climate change? A case study of large rivers in Germany, Intl Journal of Climatology, 34, 3780–3800, https://doi.org/10.1002/joc.3945, 2014.
- Kay, A.: Differences in hydrological impacts using regional climate model and nested convection-permitting model data, Climatic Change, 173, https://doi.org/10.1007/s10584-022-03405-z, 2022.
- Kay, A. L. and Davies, H. N.: Calculating potential evaporation from climate model data: A source of uncertainty for hydrological climate change impacts, J. Hydrol., 358, 221–239, https://doi.org/10.1016/j.jhydrol.2008.06.005, 2008.
- Körner, P.: Radarbasierte Niederschlagsdaten 2001 2020: Radarbasierte Niederschlagsdaten im Zeitraum 2001-2020 Beschaffung, Aufbereitung und Bereitstellung im ReKIS, https://publikationen.sachsen.de/bdb/artikel/39692/documents/60999, 2022.
- Lai, C., Chen, X., Zhong, R., and Wang, Z.: Implication of climate variable selections on the uncertainty of reference crop evapotranspiration projections propagated from climate variables projections under climate change, Agric. Water Manag., 259, 107273, https://doi.org/10.1016/j.agwat.2021.107273, 2022.
- LHW (Landesbetrieb für Hochwasserschutz und Wasserwirtschaft Sachsen-Anhalt): Bericht über das Hochwasser Januar 2011, https://klimawandel.geo.uni-halle.de/sites/default/files/abschlussbericht\_2011.pdf, 2011.
- LHW (Landesbetrieb für Hochwasserschutz und Wasserwirtschaft Sachsen-Anhalt): Bericht über das Hochwasser im Juni 2013 in Sachsen-Anhalt: Entstehung, Ablauf, Management und statistische Einordnung, last access: https://lhw.sachsen-anhalt.de/fileadmin/Bibliothek/Politik\_und\_Verwaltung/Landesbetriebe/LHW/neu\_PDF/4.0/SB\_Hochwasserschutz/Hochwasserbericht\_2013.pdf, 2014.
- Lucas-Picher, P., Argüeso, D., Brisson, E., Tramblay, Y., Berg, P., Lemonsu, A., Kotlarski, S., and Caillaud, C.: Convection-permitting modeling with regional climate models: Latest developments and next steps, WIREs Climate Change, 12, https://doi.org/10.1002/wcc.731, 2021.
- METEK: 2D Ultrasonic Anemometer, https://www.th-friedrichs.de/en/products/wind/ultrasonic-anemometer/, last access: 30 October 2023.
- Meyer, S. J., Hubbard, K. G., and Wilhite, D. A.: Estimating potential evapotranspiration: the effect of random and systematic errors, Agric For Meteorol., 46, 285–296, https://doi.org/10.1016/0168-1923(89)90032-4, 1989.
- Ndulue, E. and Ranjan, R. S.: Performance of the FAO Penman-Monteith equation under limiting conditions and fourteen reference evapotranspiration models in southern Manitoba, Theor Appl Climatol, 143, 1285–1298, https://doi.org/10.1007/s00704-020-03505-9, 2021.
- Poncet, N., Lucas-Picher, P., Tramblay, Y., Thirel, G., Vergara, H., Gourley, J., and Alias, A.: Does a convection-permitting regional climate model bring new perspectives on the projection of Mediterranean floods?, Nat. Hazards Earth Syst. Sci., 24, 1163–1183, 2024.

- Ross, C. A., Ali, G. A., Spence, C., and Courchesne, F.: Evaluating the Ubiquity of Thresholds in Rainfall-Runoff Response Across Contrasting Environments, Water Resour. Res., 57, https://doi.org/10.1029/2020WR027498, 2021.
- Rudd, A. C., Kay, A. L., Wells, S. C., Aldridge, T., Cole, S. J., Kendon, E. J., and Stewart, E. J.: Investigating potential future changes in surface water flooding hazard and impact, Hydrol. Process., 34, 139–149, https://doi.org/10.1002/hyp.13572, 2020.
- Schaller, N., Sillmann, J., Müller, M., Haarsma, R., Hazeleger, W., Hegdahl, T. J., Kelder, T., van den Oord, G., Weerts, A., and Whan, K.: The role of spatial and temporal model resolution in a flood event storyline approach in western Norway, Weather Clim. Extrem., 29, 100259, https://doi.org/10.1016/j.wace.2020.100259, 2020.
- Schulla, J.: Model Description WaSiM: Water balance Simulation Model, 2021.
- Scott, D. W.: Sturges' rule, WIREs Computational Stats, 1, 303–306, https://doi.org/10.1002/wics.35, 2009.
- Strandberg, G. and Lind, P.: The importance of horizontal model resolution on simulated precipitation in Europe from global to regional models, Weather Clim. Dynam., 2, 181–204, https://doi.org/10.5194/wcd-2-181-2021, 2021.
- Tamm, O., Kokkonen, T., Warsta, L., Dubovik, M., and Koivusalo, H.: Modelling urban stormwater management changes using SWMM and convection-permitting climate simulations in cold areas, J. Hydrol., 622, 129656, https://doi.org/10.1016/j.jhydrol.2023.129656, 2023.
- Tang, W. and Carey, S. K.: HydRun: A MATLAB toolbox for rainfall-runoff analysis, Hydrol. Process., 31, 2670–2682, https://doi.org/10.1002/hyp.11185, 2017.
- Winterrath, T., Rosenow, W., and Weigl, E.: On the DWD quantitative precipitation analysis and nowcasting system for real-time application in German flood risk management, Weather Radar and Hydrology, 323-329, 2012.

---

## Author Comment (AC2)

**Review of**

Do convection-permitting regional climate models have added value for hydroclimatic simulations? A test case over small and medium-sized catchments in Germany

By Oakley Wagner, Verena Maleska, and Laurens M. Bouwer

This manuscript presents an evaluation of the convection-permitting regional climate model ICON-CLM 2.6.4 at 3 km resolution (ICON3km) compared to its driving model at 11 km with parameterized convection (ICON11km), focusing on the Weiße Elster basin in East Central Germany. The study assesses the ability of both models to reproduce key meteorological variables (air temperature, radiation, humidity, wind speed, and precipitation) and evaluates their suitability for hydrological impact modeling using the distributed hydrological model WaSiM.

The analysis is performed for a 10-year period, from 2005 to 2014, focusing on verifying different atmospheric variables and the capability of the atmospheric models to drive a hydrological model over a series of small or medium-size catchments.

A comparison between the two models and against observed data from different sources is carried on, with the final aim to address the potential added value of the convection-permitting model.

The manuscript is well written and structured, presents new data and in general deserves to be published, although some revisions are needed to ensure that the results are better substantiated with figures/tables and that the conclusions are fully supported by the presented evidence.

**General comments:**

The abstract and introduction are clear, well-structured, and scientifically sound. They effectively present the study objectives, methods, and key results.

**Section 2.**

A more detailed discussion of the limited time period considered for a climatological analysis needs to be included in the manuscript, in particular in view of the fact that several results are characterized by statistically not significant differences between model's results.

Thank you for the comment. We will expand the manuscript by a discussion on the limitations posed by the short time series.

While subsection 2.2 (regarding the observational data is well described and comprehensive), Section 2.3 (Climate data model) should be expanded. I would suggest adding some more information on the ICON model, of the model setup and of the main parametrizations used.

We will follow this recommendation in the revision of the manuscript.

Apart from precipitation, the other atmospheric variables are verified against observations from the nearest ground station using the closest model grid value, which can be particularly problematic for temperature. In complex terrain, the altitude of the model grid can differ significantly from that of the station.

I believe it is worth adding some comments on this aspect in the subsection 2.4 and/or in the results section in the discussion of the biases, in particular for temperature.

We looked into the effect that the mismatch of the station elevations and the elevations of the associated climate model grid cells has on the temperature estimations. As a digital elevation model, we used SRTM 1 Arc-Second Global provided by USGS at 30 m resolution. We upscaled in a two-step-process, once to 1 km keeping the native grid orientation, and then to 3 km, resp. 11 km while regridding to the ICON3km, resp. ICON11km grid. We employed the environmental lapse rate  $(0.65 \, ^{\circ}\text{C}/100 \, \text{m})$  and found elevation-induced biases in the range of  $\pm 1 \, \text{K}$  (see Fig. 1 below). A set of stations, particularly in the west of the study area, are located in river valleys and are therefore at lower elevation than the climate model cells. These deviations in elevation appear as a cold bias by the climate models. Other stations are located at hill tops and therefore at higher elevation than the climate model cells, which appears as a warm bias by the climate models. We will discuss this in the revised paper.

Fig. 1: Air temperature biases induced by the offset between mean grid cell elevation and station elevation for the 20 stations in the study area (dots), plotted over a digital elevation model of 30 m resolution (SRTM 1 Arc-Second Global by USGS)

**Section 3.**

Overall, this section provides a useful comparison of ICON3km and ICON11km, but the analysis is uneven across variables. While the analysis of wind speed and precipitation is well-supported and convincing, the sections on air temperature, global radiation and relative humidity require additional figures.

Several results are not supported by any figures/table, which in my opinion should be added at least as supplementary material.

Thank you for the comment. At the core of the paper is the study of the convection-permitting regional climate model for hydrological impact modelling. To ensure that there is a balanced focus on the climatological and hydrological analyses, we can only show a selection of plots in the main body of the paper. We will however take the comment into account and include additional supporting figures in the appendix, as outlined below.

In Section 3.1.1 (Temperature), several detailed results are discussed (frequency distributions, diurnal cycle, seasonal variability of biases, DJF vs JJA differences). However, only the monthly mean biases are shown. The absence of a figure for the diurnal cycle or the frequency distribution makes it difficult for the reader to evaluate the stated findings and much of the text remains purely descriptive without visual support.

The diurnal cycles of air temperature for the four seasons as calculated by ICON11km and ICON3km and as observed are shown in Fig. 2 below.

Fig. 2: Seasonal diurnal cycles of mean hourly air temperature computed by ICON11km and ICON3km, as well as the observations for the time period of 2005 to 2014

The frequency polygons for hourly air temperature as gained from ICON11km, ICON3km and observations are shown in Fig. 3 below.

Fig. 3: Frequency polygons for hourly air temperature as calculated by ICON11km and ICON3km and as observed for the period of 2005 to 2014

We will add these figures to the appendix of the revised paper.

Similarly, in Section 3.1.2 (Global radiation) the only figure presented relates to the diurnal cycle, claims about the frequency distribution of daily mean global radiation and the monthly bias (e.g., the July improvement of 2.5 J/cm² for ICON 3km) are not supported by any figure or table . Without such evidence, this part remains insufficiently substantiated.

Fig. 4: Frequency polygons for daily mean global radiation as calculated by ICON11km and ICON3km and as observed for the period of 2005 to 2014

The annual variability of the monthly mean bias of daily mean global radiation is shown in Fig. 5.

the period of 2005 to 2014

We will expand the supplementary material by these graphs.

As regards the diurnal cycle, Figure 3 shows a one-hour shift between the models and observations throughout the entire diurnal cycle (not only for the peak) and for all seasons.

This perhaps deserves further investigation, or at least a verification of potential data misalignment, if this has not already been done.

Thank you for the comment. The climate models use the proleptic gregorian calendar and the unit of the dimension time is days since 2004-07-01T00:00:00**Z** for ICON3km and days since 1979-01-01T00:00:00Z for ICON11km. The time format is conform to ISO-8601, with **Z** indicating zero UTC offset. It is therefore in keeping with the observational data used, which is also in UTC. The plot of the diurnal cycle of temperature (c.f. Fig. 2 above) does not show a one-hour-shift, which suggests a correct representation of the climate model data. We did not identify a possible cause of the one-hour shift in the interpretation of the observational global radiation data and its peak around noon is plausible. Hence, we conclude that the temporal shift must indeed be intrinsic to the climate models.

Section 3.1.3 (Relative Humidity), similar to the previous section regarding temperature and global radiation, describes frequency distributions, monthly biases, and relative model performance (ICON3km vs ICON11km), but no figures are provided. As a result, the reader cannot verify whether the reported differences are meaningful or fall within observational uncertainty.

The frequency distribution of hourly relative humidity is shown in Fig. 6 below, the monthly mean biases in Fig. 7, and the QQ-plots as a reflection of model performance in Fig. 8. We will add the plots to the appendix of the revised paper.

Fig. 6: Frequency polygons for hourly relative humidity as calculated by ICON11km and ICON3km and as observed for the period of 2005 to 2014

Fig. 7: Monthly mean bias of hourly relative humidity of ICON3km and ICON11km for the period of 2005 to 2014, as well as the 95%-confidence intervals from the station means

Fig. 8: QQ-plots for hourly relative humidity for ICON11km and ICON3km to the point-based observations for the period of 2005 to 2014

**Section 4**

This part will probably need a slight revision after the revision of section 3.

We will adjust our discussion section (Section 4) accordingly.

**Section 5**

Overall, the conclusions are well written, clear, and consistent with the results presented in the manuscript. The authors provide a balanced discussion of both strengths (e.g., improvements in summer temperature, radiation, and wind speed representation) and limitations (notably the overestimation of heavy rainfall and its implications for discharge modelling).

**Specific comments:**

1145: The bandwidth of temperature is not clear.

Thank you, we will rewrite the temperature measurement uncertainty as  $\pm 0.08$  K to  $\pm 0.76$  K.

1150-153: The sentence is too long and could be split for clarity.

We suggest the following rephrasing: "Monthly mean biases of hourly air temperature of ICON11km and ICON3km differ only slightly (Fig. 2) and these deviations fall within the uncertainty bandwidth of the observations. Nevertheless, the results hint towards two points of improvement by the CPRCM. Firstly, ICON3km most strongly reduces the bias of its driving model ICON11km in the summer months (JJA). Secondly, the spatial bias variability throughout the year is lower for ICON3km than for ICON11km, as reflected by the 95% confidence interval from the station means."

1160-163: The sentence is too long and could be split for clarity.

We will split the sentence as follows: "Both ICON3km and ICON11km compute daily mean global radiation significantly higher than observed according to a Welch two sample t-test  $(G_{\text{ICON3km}} = 45.9 \text{ J/cm}^2, G_{\text{ICON11km}} = 45.9 \text{ J/cm}^2, G_{\text{Obs}} = 44.5 \text{ J/cm}^2)$ . However, if it is assumed that the observations are systematically too low by 3%, which is also the pyranometers' measurement uncertainty (DWD, n.d.), the apparent overestimation by the climate models is not significant any more."

I192: The sentence 'Overall, for most months ICON 3km was found to outperform ..' seems inconsistent with the preceding part of the paragraph which states that "ICON3km does not seem to offer noticeable improvement in the frequency distribution" (I190-191) and "neither of the climate models shows significant difference to the observations.

Thank you for the comment. Indeed, no improvement is visible in the frequency distribution of hourly relative humidity (see Fig. 6 above). However, results suggest lower monthly mean bias in the estimation of hourly relative humidity by ICON3km than by ICON11km for most months of the year (see Fig. 7 above). It should however be noted that these improvements are small and fall within the measurement uncertainty bandwidth.

We will clarify this in the revised paper and include the plots as suggested in the comment to section 3.1.3. We intend to modify the corresponding paragraph as follows: "For most months of the year, monthly mean bias of hourly relative humidity is lower with ICON3km than with ICON11km. Only in the months of spring (MAM) was ICON11km found to outperform ICON3km. However, it should be noted that the largest absolute

difference in the monthly means of hourly errors between the two models occurs in September and October and is of only 1.3 % relative humidity. These deviations fall within the measurement uncertainty bandwidth."

1247: I suggest simplifying the phrase "in the summer month of July" to just "in July".

Thank you. We will follow the recommendation.

1445-448: The sentence is too long and could be split for clarity.

We suggest to split the sentence as follows: "In the work presented here, the skill of estimating meteorological variables on the catchment scale is analysed for ICON-CLM 2.6.4 in its convection-permitting setup at 3 km resolution (ICON3km), and for its driving model ICON-CLM 2.6.4 at 11 km resolution with parametrised convection (ICON11km, forced with ECMWF-ERA5). Analyses are conducted exemplarily over the Weiße Elster basin in East Central Germany for the historical period of 2005 to 2014."

---

## Author Comment (AC3)

**Review of**

Do convection-permitting regional climate models have added value for hydroclimatic simulations? A test case over small and medium-sized catchments in Germany

By Oakley Wagner, Verena Maleska, and Laurens M. Bouwer

**General comments**

This paper compares the efficiency of a Convection-Permitting Model (CPM: ICON-CLM 2.6.4 at 3km) and its driving Regional Climate Model (RCM: ICON-CLM 2.6.4 at 11km) to simulate some climate variables and discharge when forcing WaSim distributed hydrological model. The study is conducted over a cluster of catchments of small and medium size in central eastern Germany. The evaluated climate variables are the one relevant for hydrological modelling such as surface temperature, relative humidity, wind speed, global radiation and precipitation. The CPM exhibits no clear added-value for the simulation of all the climate variables. For precipitation, especially extreme precipitation, the increase in model resolution switches from a negative bias with the RCM to a strong positive bias and sometimes unrealistic values with the CPM. This behavior leads to an overestimation of discharge over the studied catchments even if the strongest recorded flood of the period remains underestimated.

Due to the scarcity of hydroclimatic studies using high resolution regional climate models, this paper is welcomed in the community, even if it consists of a simple case study using only one CPM-RCM couple and one hydrological model on one particular cluster of catchments. The study adds further scientific content in the CPM bibliography by assessing a CPM potential benefits and understanding their transferability to the hydrology.

The article is interesting, well organized and globally well written. However, some important information is lacking and some important aspects need to be addressed to improve either the clarity of the message and the consistency of the paper to make it suitable for publication in HESS.

1 - According to the paper title, the aim of the study is the assessment of the added value of the CPM compared to the RCM on hydroclimatic simulations, meaning climate and related hydrological simulations. The evaluation of global radiation, wind speed and relative humidity, even if bringing interest, are too detailed for the real purpose of the paper. The evaluation results presented for those variables are discussed separately and are not connected in any way to the final output that is the hydrological simulations.

If keeping this level of details (annual cycles of global radiation, diurnal cycles of wind speed, relative humidity) I would expect to see :

 a) A quick comparison of Potential Evapotranspiration (PE) obtained from simulations and observations to assess the impact of these variables on hydrological model forcing. PE could be directly accessible from the hydrological model output. If not, you could compute it with the Penman-Monteith equation from the climate data. PE annual and diurnal cycles would be at least as interesting and relevant for your study as diurnal cycles of wind speed and annual cycles of global radiation.

The monthly mean bias of hourly total evapotranspiration (ET), as calculated by the hydrological model driven with ICON3km and ICON11km in respect to when driven with observational data (with precipitation data from RADOLAN3km and RADOLAN11km respectively), is shown in Fig. 1. During summer, the biases of the climate models differ most strongly. In fact, the identified reduction of the negative summer bias in temperature and global radiation by ICON3km may contribute to the (slightly) higher ET estimates in summer. We will expand the paper by analyses of simulated evapotranspiration.

Fig. 1: Monthly mean bias of hourly total evapotranspiration, as calculated by the hydrological model driven with ICON3km and ICON11km for the period 2006 to 2014

• b) If possible, the impact on these variables on snowmelt in the model, and their contribution to discharge biases.

Over the nine-year simulation period, the hydrological model driven with meteorological data from the climate models overestimates the catchment spatial average nine-year snow melt sum (1538 mm for ICON11km compared to 1103 mm for RADOLAN11km, and 1571 mm for ICON3km compared to 1113 mm for RADOLAN3km). This is reflected in a positive discharge bias in early spring. The focus of this paper is however on convective summer storms, where CPRCMs and RCMs are likely to show the greatest differences.

2 - Some important results of the paper are discussed but never presented in tables or figures. On the contrary some others are, in my opinion, secondary if the analysis proposed in the previous comment is not produced. These intermediate results take up space at the detriment of the hydrological analysis which is the main point of the study.

• a) I308-311 and 426-428: You present and discuss results concerning discharges above the 99.5th percentile but these results are never shown in tables or figures. Yet, one of the main reasons for CPM development and their use is to enhance impact studies for extreme events such as floods. Here, the results about extreme discharges are really important to show, either under the form of CDF comparison, QQ plots or even a table comparing biases for particular quantiles of discharge.

Boxplots of the 99.5th percentile of hourly discharge computed by the hydrological model driven with ICON3km, ICON11km, RADOLAN3km and RADOLAN11km respectively are displayed in Fig. 2 and will be added to the appendix of the revised paper.

Fig. 2: Boxplots of the 99.5h percentiles of hourly discharge (period of 2006 to 2014) computed by the WaSiM hydrological model driven with meteorological data from ICON11km and ICON3km, as well as with adjusted radar data of respective equal resolution (RADOLAN11km and RADOLAN3km) for catchments of the main stem of the Weiße Elster river within the study area. Boxplots are built according to McGill et al. (1978).

• b) Too much attention is brought to wind speed frequency evaluation, for really tiny frequency biases that probably do not affect discharge modelling at all. In my opinion, seeing intensity biases on the whole distributions would be more informative. Figure 4 should be moved to supportive information and potentially replaced by a QQ plot of the type shown in figure 6.

QQ-plots for hourly wind speed over the period of 2005 to 2014 for ICON11km and ICON3km to the point-based observations are shown in Fig. 3. We will add the figure to the appendix of the manuscript.

Fig. 3: QQ-plots for hourly wind speed over the period of 2005 to 2014 for ICON11km and ICON3km to the point-based observations

• c) I221 - 225: The main finding in Figure 6 (and later Figure 8) that stands out is the significant overestimation of precipitation for ICON3km, with unrealistic values (> 100mm in 1 hour). Either here or in the discussion, these surprising results are not highlighted, nor discussed enough. What are the synoptic scale conditions leading to the occurrence of these values? Where do they occur? Are they numerical errors? Are these findings in line with previous versions of ICON3km?

We will conduct event analyses with a focus on the registered very high rainfall intensities to detect among others if they group together and to determine their seasonality.

Furthermore, the median of precipitation above the 99.5th percentile is not representative
of extreme values. An option is to conduct the same computation over a different quantiles
(25th percentile, median, 90th percentile, 95th percentile and 99th percentile) of wet
hourly rainfall (> 0.1mm/h or >1mm/h) distribution. The biases of ICON3km seem to exceed
200% for the highest quantiles of wet hourly precipitation.

The 99.5th percentile is by definition a reflection of the extreme values and we present this analysis therefore to discuss the heavy precipitation simulations.

3 - Figure 1 is a map showing the study area and its contrasted topography. In the rest of the text, the biases of climate variables are presented either aggregated on the whole area or without

distinction of geographical location. As a case study over a limited area, it would be very interesting to use the distributed nature of the data to visualize certain results in a map format. For example, a map of extreme precipitation biases for the two climate models, even in their native resolution, would be of great help to understand the spatial distribution of these biases and their transfer role in hydrological biases. It could as well help to understand the spatial patterns of hourly extreme rainfall values (> 100mm/h, visible in figure 6). Are these values simulated on high elevation areas (potentially related to biases of the orography forcing) or randomly over the study area (numerical errors)?

This map could be included in the main body or in the supportive information depending on the relevance and of the article available space.

We employ a two-nest approach, in which the regional climate model ICON-CLM is run over the EURO-CORDEX domain at 11 km resolution driven by ECMWF-ERA5, and ICON-CLM in convection-permitting setup at 3 km resolution is embedded over the Central European (CEU) domain. With domains of such size and without nudging, high-intensity rainfall events are unlikely to be simulated at the precise location they have been observed. As such, a pixel-based computation of bias and its map-representation are not advised.

We will conduct event-based analyses and look at the most extreme storms simulated, mapping the climate model results side-by-side.

- 4 The methodology section should be complemented by:
- a) A small paragraph presenting the hydro-climatic conditions of the study area (climate type, hydrological regime...)

Thank you for the comment. We will implement it.

b) A new paragraph presenting hydrological model calibration and validation periods and choices. We understand later on in the results (I 269-270 and I275-276) that the authors chose to calibrate WaSim on the June 2013 flood and validate it over a winter flood, but no periods or dates are specified. This methodology remains therefore unclear and arises some questions:

The analysis of the paper focuses on the whole range of discharges, but not
particularly on extremes (except for one particular flood). Why have you chosen to
calibrate the hydrological model only on extreme events?

The hydrological model was calibrated on the 2013 flood, as well as on the calendar year of 2012. We looked at weekly discharge sums, whereby the weeks of the year were defined according to ISO 8601.

What are the exact period dates for calibration and validation?

The periods for calibration are:

- the 2013 flood event: 2013-05-31 00:00 to 2013-06-06 23:00
- the 2012 calendar year: 2012-01-02 00:00 to 2012-12-30 23:00

The periods for validation are:

- the 2011 flood event: 2011-01-07 00:00 to 2011-01-24 23:00
- the 2017 calendar year: 2007-12-31 00:00 to 2008-12-28 23:00
- Can certain hydrological processes, such as those governing long-term soil moisture, be excluded from calibration and therefore poorly simulated?

Long-term processes should receive consideration in calibration, therefore we have also calibrated our model on a complete calendar year.

• A classical modelling approach is to perform a split-sample test to calibrate and validate your hydrological model (KLEMEŠ, 1986). Given the short period of data, another recommended approach is to calibrate over the entire length of data (Arsenault et al., 2018; Shen et al., 2022). Have you tried one or the other approach? If so, it would be interesting to see a summary of the calibration/validation results. If not, could you justify your calibration choices?

We calibrated the hydrological model on the most severe summer flood observed in the catchment, as convective summer storms are particularly interesting in the discourse on the added value of CPRCMs for hydrological impact modelling. In fact, we showed that biases of ICON3km and ICON11km deviate most strongly during the summer months, likely since deep convection is parametrised in ICON11km and explicitly resolved in ICON3km.

We furthermore calibrated the model on the calendar year of 2012, a year of comparatively low monthly precipitation anomalies.

The paragraph should clarify this aspect of methodology, by justifying the authors' choices and answering the question above as best as possible.

**Specific comments**

• In my opinion, the passive form is overused and makes the text sometimes difficult to read. The expression "was/were found to" is too frequent (lines 154, 170, 175, 188, 192, 193, 204, 211, 250, 258, 260, 274, 276, 283, 297, 306, 332, 341, 346, 358, 366, 369, 378, 392, 396, 403) and could be changed in many cases by a more active form. For example: "ICON3km was found to overestimate" could be replaced by "ICON3km tends to...", "ICON3km shows an overestimation" or "ICON3km overestimates...", or "We notice an overestimation..."

Thank you for the suggestion. The phrasing has been chosen deliberately to underline that the conclusions were drawn from the one simulation run we did. The active form (e.g. "ICON3km overestimates") entails the risk of assumed generalisation, such as that ICON-CLM per se strongly overestimates rainfall intensities. We will revise the text for improved readability.

• Table 1: The table 1 in Introduction listing existing hydroclimatic studies using CPM is greatly appreciated, but needs to be completed with the most recent work up to date (Dale and Shelton, 2025; Xie et al., 2025, maybe others...).

Thank you very much for pointing this out. We will update the literature review.

• I47: This sentence is misleading. The part of the sentence "come to offer substantial added value for flood simulation" implies that a consensus has been reached about the added-value of CPM on discharge modelling, which is not the case yet, thus justifying the interest of your study. The listed studies in Table 1 focus on various aspects of hydrological modelling, from low flows to floods, covering very different regions and climates and using diverse CPM and are too few to conclude. I would recommend the authors to take into account this comment and change this part of the sentence.

We will rewrite the respective sentence taking the comment into account.

• 181: To compare point-based climate data to stations, are you applying a correction for the temperature vertical profile depending on grid mean elevation and station elevation?

Thank you for the comment. We did not apply a correction for the temperature vertical profile. We have followed this up and taken a look at the effect that the mismatch of the station elevations and the elevations of the associated climate model grid cells has on the temperature estimations. As a digital elevation model, we used SRTM 1 Arc-Second Global provided by USGS at 30 m resolution. We upscaled in a two-step-process, once to 1 km keeping the native grid orientation, and then to 3 km, resp. 11 km while regridding to the ICON3km, resp. ICON11km grid. We employed the environmental lapse rate (0.65 °C/ 100 m) and found elevation-induced biases in the range of  $\pm$  1 K (see Fig. 4 below). A set of stations, particularly in the west of the study area, are located in river valleys and therefore at lower elevation than the climate model cells. These deviations in elevation appear as a cold bias by the climate models. Other stations are located at hill tops and therefore at higher elevation than the climate model cells, which appears as a warm bias by the climate models. We will discuss this in the revised paper.

Fig. 4: Air temperature biases induced by the offset between mean grid cell elevation and station elevation for the 20 stations in the study area (dots), plotted over a digital elevation model of 30 m resolution (SRTM 1 Arc-Second Global by USGS)

• I111: If I understand this part of methodology well, precipitation observations have been upscaled to the climate model resolution, only with an aggregation of the initial grid and no regridding? No interpolation has been done?

The RADOLAN product has been retrieved at 1 km resolution. It has been upscaled and regridded to the ICON3km and ICON11km grid (becoming RADOLAN3km and RADOLAN11km).

I113-115: This part of the sentence has to be rephrased to improve clarity.

We will do that.

• I125 : Can you justify the differences of duration increments between ICON3km and ICON11km?

Sliding a short moving window over an hourly time series of 10 years to extract the precipitation sums within each window position is a computationally intensive endeavour. For the 11-km-resolution data, we worked with window sizes of 2 h, 4 h to 24 h by 2-hour-increments. For the higher resolution data (3 km), we increased the increment size from 2 hours to 4 hours to stay within a reasonable computation time.

• In section 2.4 or 3.1, please explicitly detail the extent of the evaluation domain. Is it the rectangle displayed in figure 1.b or the mask of the catchments?

Thank you for identifying this lack of clarity. The evaluation domain is the whole study area, i.e. the rectangle displayed in figure 1.b in the manuscript. We will clarify this in the revised paper.

• I146-147 and 148-149: These results (temperature estimates and frequency distribution) are not linked to any figure or table. Please show them.

An overview of the temperature estimates is given by the boxplots in Fig. 5 below, whereas the frequency distributions are shown in Fig. 6.

Fig. 5: Boxplots of hourly air temperature for the observations, ICON3km and ICON11km for the period of 2005 to 2014. The boxplots are built according to McGill et al. (1978).

Fig. 6: Frequency polygons for hourly air temperature as observed, and calculated by ICON3km and ICON11km for the period of 2005 to 2014

I154: Could you be more specific about the sign of the error (negative)?

Monthly mean bias of hourly air temperature by ICON11km and ICON3km was either zero or negative for all months of the year. We will add this to the text in the paper.

• I171-173: This sentence should be rephrased to clarify the message: the passive voice and all the commas make it difficult to read.

We suggest the following rephrasing: "A similar picture can be drawn for the intensity estimates of global radiation. The climate models show a positive bias in the intensity of moderate daily mean global radiation. Extremes however were underestimated."

• I176: Can you convert this 2.5J/cm2 bias reduction in percent? It is hard to realize if it is a big improvement or negligible.

A unit of absolute bias of daily mean global radiation translates to a higher relative bias on a cloudy day than on a sunny day. We therefore abstain from computing relative biases of daily mean global radiation. For reference, station-averaged daily mean global radiation in July was of 77.6 J/cm2 for the study period of 2005 to 2014. We will add this number to the paper.

• I194-195: This sentence is hardly understandable. I propose this modification: "The largest difference in the average monthly errors between the two models over the year is only 1.3%, ...". A figure could help visualize this aspect.

We will adopt the sentence as follows: "The largest absolute difference in the monthly means of hourly errors between the two models occurs in September and October and is of only 1.3 % relative humidity." Fig. 7 below supports the statement and will be added to the appendix.

Fig. 7: Monthly mean bias of hourly relative humidity of ICON3km and ICON11km during the period 2005 to 2014, as well as the 95%-confidence intervals from the station means

• 1201-202: What is the measurement uncertainty for relative humidity?

The measurement uncertainty for relative humidity is stated in line 186 of the paper. The employed sensor HMP45D comes with a measurement uncertainty of  $\pm$  2 % (Kyrouac & Theisen, 2017).

L237 : change "In keeping is" by "We notice"

Here we juxtapose the conclusions drawn from the depth-duration-frequency curves (with durations of up to 25 hours) to the yearly precipitation totals. The overestimation of high precipitation intensities and their frequencies by ICON3km is reflected both in the depth-duration-frequency curves and the yearly precipitation totals. As such, the results are in keeping but cannot be derived from each other, wherefore we chose to introduce the yearly precipitation totals using "in keeping".

L246: A presentation and an analysis of the diurnal cycle of precipitation would be
interesting, considering the task is done for global radiation and wind speed. In addition, it
could help to understand the representation of summer convective precipitation by the
model and the biases shown for July in Fig S4.. Depending on the length of the paper, this
could fit in Supportive Information.

We studied the diurnal variation of median hourly precipitation during wet days and did not find a clear signal in the climate model data. In fact, Ban et al. (2021) who show the diurnal cycles for over 20 simulations at 3 km resolution and 12 km resolution respectively over Switzerland, France and Italy, also did not see a noticeable diurnal cycle of mean precipitation over France in summer for some of the climate models. Their ensemble has a large intermodal spread, but in the mean does however show an improvement by the CPRCMs.

• L254 - 256: This sentence should be rephrased. Here is a suggestion: "Additionally, the finer resolution of ICON3km improves the delineation of heavy rainfall events, preventing runoff generation and routing outside of the concerned catchments, as it occurs in RCM"

Thank you for the suggestion, which however does not reflect what the initial sentence intended to convey. We can suggest the following alternative phrasing: "Thereby they offer great advantage for hydrological impact modelling. In fact, coarse climate model cells overlapping the catchment boundaries bear the risk of having their precipitation estimates risen by heavy rainfall that occurs just outside the catchment. As these climate model cells stretch into the catchment, their high precipitation estimates will be attributed to the catchment, when in fact the storm occurred slightly outside of the catchment."

• 1259: I suggest to cut the sentence after ICON3km and to start after it with: "This is also reflected..."

We will follow the suggestion.

• 1276-277: Could you share an example of this behavior?

The validation results are a representative example of the model's ability to simulate snow melt floods. Similar behaviour can be seen for other snow melt induced floods in the time series. Flood peaks are computed as too high and retarded. The hydrographs are depicted as too flashy, while the baseflow is underestimated. We validated our model on a snow melt flood, as no other summer flood of comparable return period is present in the time series. The focus of the paper is however on heavy convective rainfall events.

• I278 : "were too high" → "underestimates"

We can adjust the wording and substitute "the model results were too high (...)" with "the models **over**estimate (...)"

l291: The sentence of fig 10 legend is too long. Could you separate it into different parts?

Our boxplots are built according to McGill et al. (1978). We will indicate this in the methodology section and leave out the specifications in the figure captions.

I304: What do you call the "full range of ICON11km meteorological data"?

We will rewrite the corresponding sentence as: "Driving the hydrological model with ICON11km meteorological data was found to lead to an underestimation of median hourly discharge, but results in an overestimation when looking only at the 99.5th percentile of hourly discharge for six out of the seven catchments on the main stem."

• 1306: Why are you considering only catchments relative to the main stem?

For all catchments in the study area, the climate models show a similar annual course of bias. Given the similarities, we opted for a 2x2-plot-matrix and limited visualisation to the four most downstream catchments on the main stem. It should be noted that bias on upstream catchments is reflected in downstream catchments.

• 1314: In methodology, it is stated that the calibration was performed on the July 2013 flood. How did you calibrate the hydrological model on other catchments if discharge measurements are not available?

We calibrated the models using the discharge measurements at Eisenhammer, Weida, Strassberg, Elsterberg, Greiz, Gera Langenberg and Zeitz. The catchments of Kleindalzig, Berga, Dröda, Pöhl and Mylau are ungauged. However, with the sole exception of Kleindalzig, each of these catchments feeds into gauged downstream catchments and their effects are thereby seen in the downstream discharge measurements.

• 1336-340: This discussion is interesting. This behavior could have been checked easily in your study by looking if the temperature biases are more important on Tmin than Tmax.

We have followed this up by defining Tmin as temperature measurements below the 0.5th percentile and Tmax as temperature measurements above the 99.5th percentile. For the corresponding hourly time steps, we identified higher bias by the climate models for Tmin (ICON3km:  $3.3 \pm 4.2$  K, ICON11km:  $3.0 \pm 4.3$  K) than for Tmax (ICON3km:  $-1.2 \pm 2.2$  K, ICON11km:  $-1.3 \pm 2.1$  K). However, the biases on Tmin are positive and therefore not a reflection of the relatively strong negative biases we saw for winter when looking at the monthly means of hourly biases across the whole distribution. We attributed these

relatively high negative monthly mean biases to higher albedo and shorter roughness lengths on cropland and pasture in winter (Lind et al., 2020). However, extremely cold days are likely to be windstill and the presence of snow is not a given. As such, we are likely to see these effects on the monthly means of hourly biases and not on Tmin.

• 1342-344: The source of summer temperature improvement can be checked in your study by looking at the improvement of diurnal cycle of precipitation. Have you taken a look at it?

We looked at the diurnal cycle of precipitation. Please refer to our answer to the comment to L246.

• 1352-353: Wrong citation yea. It is Keller et al, 2016.

Thank you for pointing this out.

• Fig 4 and lines 366-367 in discussion: The underestimation of extremes is not clear. There is a very slight underrepresentation of very light winds (can we consider them extremes?) but no visible results on strong winds. To state this, you should represent and analyze a QQ-plot of ICON3km and ICON11km against observations, as advised in Major comment 2.b, and change these lines.

The QQ-plots for hourly wind speed for ICON11km and ICON3km to the point-based observations are presented in Fig. 3 above. The climate models show an underestimation of the intensity of high wind speeds. The figure will be added to the appendix of the revised paper.

• I380-381: This sentence should be rephrased for clarity.

We suggest alternatively: "These results agree with the consensus from literature (Lucas-Picher et al., 2021), e.g. with Ban et al. (2021) and Adinolfi et al. (2021). In their study over the southern United Kingdom, Kendon et al. (2012) showed an RCM of 12 km resolution to compute heavy rainfall as not intense enough, but as too persistent and widespread."

• I403: I advice the authors to rephrase the sentence to something like: "ICON3km does not improve the simulation of monthly precipitation, exhibiting a negative bias as ICON11km"

ICON3km exhibits a **positive** bias, just as does ICON11km. Here we discuss negative monthly precipitation anomalies, i.e. months during which the observed precipitation sums were below the long-term mean. We will clarify what we mean with negative monthly precipitation anomalies.

• I414-415 and 425-426: You have to be careful in your statement here. The different studies are not analyzing the same aspects of hydrology. Some did for the whole discharge range and others only for floods. Please modify these statements accordingly.

Thank you for pointing this out. We will refine our statement.

• I436-437: Complete the sentence here to remind that the study is a case study and the results are valid over these particular catchments with the specific used climate models.

We will clarify this in the manuscript.

Conclusion: a sentence should be added to contrast between the results and what is
written line 47. Your study is not in line with recent studies and brings an interesting result:
CPM does not systematically perform better than RCM for hydrological simulation because
of some important biases on climate variables (here extreme precipitation). However, these
results are consistent with some aspects of Xie et al. (2025) study.

We will follow this recommendation.

• 1447 : You should split the sentence at the comma and start again with "The study is conducted exemplarily..."

Thank you, we will do so.

• I461: From which study are you concluding of a "particular potential"? This paper cannot conclude on a potential of ICON-CLM given the strong biases and the absence of added-value of using this CPM for hydrological simulations. I suggest you end your conclusion on an open note on the efforts put in place to correct some biases of the CPM (bias-correction, development, better understanding on extreme precipitation biases of ICON3km) to enhance CPM simulation and impact studies.

We will extend our concluding remarks taking these comments into account.

**References**

- Ban, N., Caillaud, C., Coppola, E., Pichelli, E., Sobolowski, S., Adinolfi, M., Ahrens, B., Alias, A., Anders, I., and Bastin, S.: The first multi-model ensemble of regional climate simulations at kilometer-scale resolution, part I: evaluation of precipitation, Clim Dyn, 57, 275–302, https://doi.org/10.1007/s00382-021-05708-w, 2021.
- Kyrouac, J. and Theisen, A.: Biases of the MET Temperature and Relative Humidity Sensor (HMP45) Report, 2017.
- Lind, P., Belušić, D., Christensen, O. B., Dobler, A., Kjellström, E., Landgren, O., Lindstedt, D., Matte, D., Pedersen, R. A., and Toivonen, E.: Benefits and added value of convection-permitting climate modeling over Fenno-Scandinavia, Clim Dyn, 55, 1893–1912, https://doi.org/10.1007/s00382-020-05359-3, 2020.
- McGill, R., Tukey, J. W., and Larsen, W. A.: Variations of Box Plots, Am. Stat., 32, 12, https://doi.org/10.2307/2683468, 1978.